



## Investigating size-segregated sources of elemental composition of particulate matter in the South China Sea during the 2011 *Vasco* Cruise

Miguel Ricardo A. Hilario[a], Melliza T. Cruz[b], Maria Obiminda L. Cambaliza[a,b], Jeffrey S. Reid[c],

Peng Xian[c], James B. Simpas[a,b], Nofel D. Lagrosas[a, b, *], Sherdon Niño Y. Uy[b], Steve Cliff[d],

Yongjing Zhao[d]

[a] Department of Physics, Ateneo de Manila University, Quezon City, Philippines

[b] Manila Observatory, Ateneo de Manila University campus, Quezon City, Philippines

[c] Marine Meteorology Division, Naval Research Laboratory, Monterey, CA, USA

[d] Air Quality Research Center, University of California Davis, CA, USA

*Correspondence to*: Maria Obiminda L. Cambaliza (mcambaliza@ateneo.edu)

[*] Now with Center for Environmental Remote Sensing, Chiba University, Japan

**Abstract**

The South China Sea/West Philippine Sea (SCS/WPS) is a receptor of various natural and anthropogenic aerosol species from throughout greater Asia. In combination with its archipelagic/peninsular terrain and strong Asian monsoon climate, the SCS/WPS hosts one of the most complex aerosol-meteorological systems in the world. However, aside from the well-known biomass burning emissions from Indonesia and Borneo, the current understanding of aerosol sources is limited-especially in remote marine environments. In September 2011, a 2-week research cruise was conducted near Palawan, Philippines to sample the remote SCS/WPS environment. Size-segregated aerosol data was collected using a Davis Rotating-drum Unit size-cut Monitor sampler and analyzed for concentrations of 28 selected elements. Positive Matrix Factorization (PMF) was performed separately on the coarse, fine, and ultrafine size ranges to determine possible sources and their contributions to the total particulate matter mass. Additionally, size distribution plots, time series plots, back trajectories and satellite data were used in interpreting factors. Using tracers of various sources, a linear regression analysis and correlation matrices showed the presence of soil dust and sea spray in the coarse mode, biomass burning in the fine mode and oil combustion in the ultrafine mode. Mass distributions showed elevated aerosol concentrations towards the end of the sampling





period which coincided with a shift of air mass back trajectories to Southern Kalimantan. Covariance between coarse and fine
mode sources were observed. The PMF analysis resolved five sources across the three size ranges: biomass burning, oil
combustion, soil dust, sea spray and a fly ash factor largely composed of heavy metals. The agreement between the PMF and
the linear regression analyses suggests the robustness of the PMF solution. While biomass burning is indeed a key source of
aerosol, the study shows the presence of other important sources in the SCS/WPS. Understanding these sources is key to
characterizing the chemical profile of the SCS/WPS and, by extension, developing our understanding of aerosol-cloud behavior
in the region.

## 1.    Introduction

The South China Sea/West Philippine Sea (SCS/WPS) is a receptor of a multitude of natural and anthropogenic

aerosol species.  At the same time, due to its archipelagic/peninsular terrain coupled with a strong Asian monsoon climate, the
region exhibits some of the world's most complicated meteorology.  Together, the SCS/WPS hosts one of the world's most
complex and sensitive composition and climate regimes (Balasubramanian et al., 2003; Yusef and Francisco, 2009; Atwood
et al., 2013; Reid et al., 2012, 2013, 2015). Particles in the atmosphere are known to influence radiative forcing via absorption
and scattering of solar radiation (Nakajima et al., 2007; Boucher et al., 2013; Lin et al., 2013; Ge et al., 2014) and act as cloud
condensation nuclei (CCN), affecting cloud reflectivity, evaporation and precipitation rates (Sorooshian et al., 2009; Lee et
al., 2012; Boucher et al., 2013; Ross et al., 2018). The northern portion of the SCS/WPS has been known to be impacted by
not only China via dust storms (Wang et al., 2011; Atwood et al., 2012) and industrial pollution but also by Southeast Asia
through its own anthropogenic pollution and biomass burning (Lin et al., 2007; Cohen et al., 2010a, b; Wang et al., 2011; Reid
et al., 2015, 2016). Countries surrounding the Maritime Continent (MC) are known to be impacted by strong seasonal burning
(Balasubramanian et al., 2003; Reid et al., 2013). The atmospheric residence times of fine particles allow for long range
transport, potentially creating regional and global concerns (Cohen et al., 2010a).

Highlighting the unique combination of terrain and sea that feeds into the complexity of the meteorological

environment of the region, Reid et al. (2012) and Xian et al. (2013) posed the long-range hypothesis that monsoonal flows and
higher-frequency meteorological phenomena are a major factor in seasonal aerosol dispersion. Biomass burning plumes are
known to cause severe haze episodes due to these monsoonal flows, raising concentrations of particulate matter (PM) to impact
cloud physics and, in some cases, to dangerous air quality levels across large areas, particularly in association with positive
phases of the El Niño-Southern Oscillation (ENSO) (Engling et al., 2014; Fujii et al., 2015). Likewise, biomass burning is a
significant contributor to the region's CCN budget in all years as are the region's significant anthropogenic emissions
(Balasubramanian et al., 2003; Field et al., 2008; Reid et al., 2012; 2013; 2015; 2016; Atwood et al., 2017).

Partly due to the emphasis on dramatic biomass burning as the primary source of aerosol particles in the region, the

contributions of other regional sources are not well understood or perhaps underappreciated. As the SCS/WPS is host to major
population centres, industry, major ports, and coal and oil combustion are expected to be an important regional source of




aerosol particles in the MC. Soil dust and coarse mode biological particles may also play a role in as ice nuclei (O'Sullivan et
al., 2014), as biomass burning plumes are known to entrain such particles (Reid et al., 1998; 2005; Schlosser et al., 2017). As
such, a network of interacting sources exists in the region surrounding the SCS/WPS, wherein sources mix during transport
and complicate source apportionment. Understanding the nature of sources in the remote MC and their contributions is key to
characterizing the aerosol environment in the SCS/WPS and its relationship with cloud behavior and precipitation patterns in
the region; this is particularly true given the higher sensitivity of clouds to particle perturbations at lower concentrations.
However, the identification of sources is complicated by their complex chemistry and interactions with the marine environment
(Atwood et al., 2012; 2017).

As part of the Seven South East Asian Studies program (7-SEAS), a research cruise (Reid et al., 2015) was conducted

in late September 2011 onboard the Philippine-flagged M/Y *Vasco* in the vicinity of the northern Palawan archipelago. The
goal of this cruise was to observe the behavior of aerosol particles in the SCS/WPS and test the transport hypothesis proposed
in Reid et al. (2012) that the Philippines is a long-range receptor of aerosol species transported across the SCS/WPS during
the boreal summer southwest monsoon from Borneo, Sumatra, and the Malay Peninsula. In particular, the cruise aimed to
observe that MC emissions were reaching the Southwest Monsoon monsoonal trough. The Palawan archipelago is a good
receptor site for regional emissions due to its largely rural settlements and its location upwind relative to the rest of the
Philippines. The sampling period coincided with the passage of one tropical storm and two tropical cyclones (TC). Of particular
importance is the passage of a supertyphoon Nesat beginning on 26 September as TC inflow arms are known to cause abrupt
changes in regional flows.

As part of the 2011 *Vasco* cruise particulate matter was collected using a size segregated Davis-Rotating Uniform

Size-Cut Monitor (DRUM) impactor analyzed for elemental composition. While Reid et al. (2015) noted the presence of
plumes in two episodes during the cruise, their initial analysis of the region's atmospheric chemistry also suggested the events
were a mix of biomass burning and oil or shipping emissions due to elevated levels of vanadium. Additionally, differences in
elemental ratios, mass fractions and back trajectory origins between the two events support the presence of other sources
besides biomass burning. From the initial analysis of aerosol chemistry presented by Reid et al. (2015), this study aims to
investigate aerosol sources in the SCS/WPS and to further develop the current understanding of the effect of regional
meteorological phenomena on aerosol dispersion. The paper shows that, though biomass burning is a major source of aerosols
in the SCS, anthropogenic sources such as oil combustion also play an important role in the chemical profile of the region. As
we report, soil transport was observed as well.

In this paper we expand on the original 2011 *Vasco* cruise analysis to quantitatively apportion sampled biomass

burning and anthropogenic aerosol species. Positive Matrix Factorization (PMF) was performed on size-segregated PM to
detect possible size-specific sources (Han et al., 2006; van Pinxteren et al., 2016). Indeed, the relationship between the
aerodynamic diameter of a particle and its source has been well-established in literature (Reid et al., 1993; Balasubramanian


et al., 2003; Han et al., 2006; Lestari et al., 2009; Wimolwattanapun et al., 2010; Santoso et al., 2010; Karanisiou et al., 2009;
Seneviratne et al., 2010; Atwood et al., 2012; Lin et al., 2015; Cahill et al., 2016).  Aerosol factors and characteristics were
then used to spawn back trajectories to identify individual island emissions areas.
**2. Sampling and Methods**
**2.1. Overall cruise sampling and environment**

A general overview of the 2011 cruise can be found in Reid et al. (2015) and a brief summary is provided here.

Sampling was conducted around the Palawan archipelago, an island chain located at the southwestern edge of the Philippines
in between the SCS/WPS and the Sulu Sea. Sampling was performed between Manila and the northern tip of Palawan Island
onboard the M/Y *Vasco* which left Manila Bay on 17 September 2011 and returned on 30 September 2011 (Fig. 1). Majority
of samples were collected around the areas of El Nido and Malampaya Sound (111.1° N, 119.3° E) where the vessel was on
station from 21-28 Sept. The largely rural population of Palawan made it an ideal receptor for regional rather than local
emissions.

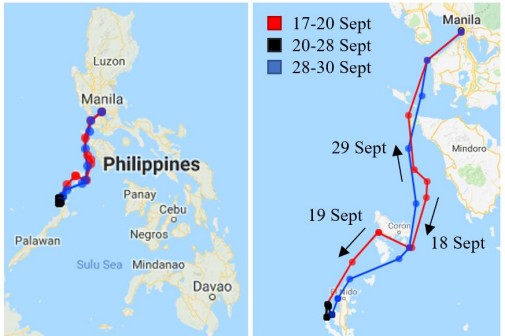


**Figure 1. Path taken by the M/Y *Vasco* for 17-20 September (red), 20-28 September (black), 28-30 Sept (blue). Majority**
**of sampling was done at the northern end of Palawan island. Image courtesy of GoogleMaps.**

The cruise was conducted at the end of the boreal summer monsoon which usually lasts from June through September

(Loo et al., 2014; Chang et al., 2005). The Asian monsoon is caused by the annual march of the sun and asymmetrical heating
of air masses due to the complex terrain of Southeast Asia (Chang et al., 2005). The campaign coincided with the peak burning
season in Southern Kalimantan and Southern Sumatra, which have been measured to be the highest emitters of biomass burning
plumes in the MC (Reid et al., 2012). As the southwest monsoon is characterized by winds travelling southwest to northeast,
Reid et al. (2015) proposed that the Philippines was an excellent receptor for regional emissions from the MC.

Although 2011 was a moderate La-Niña year, it was noted that fire activity and precipitation levels resembled a

neutral year (Reid et al., 2015). The cruise took place when the Madden-Julian Oscillation (MJO) was transitioning from the





wet phase to the dry phase, which is expected to enhance burning activity and transport. With the passage of tropical cyclones
(TCs), significant aerosol events were observed to propagate across the region.
Reid et al. (2015) described three tropical events that occurred during the cruise, specifically tropical storm (TS)
Haitang, super-TC Nesat, and super-TC Nalgae. The presence of inflow arms in the SCS has been suggested to affect the
aerosol environment by bringing more MC air into the region (Reid et al., 2015). The passage of Nesat was observed to abruptly
affect air mass trajectories coinciding with an enhancement of several elements during the last two days of the cruise.
Figure 2 shows the evolution of the meteorological environment over the cruise period with comparisons between
satellite-derived aerosol optical depth (AOD) derived from the MODIS-Terra and MODIS-Aqua satellites, back trajectories
from NOAA Hybrid Single Particle Lagrangian Integrated Trajectory Model (HYSPLIT) and 850 hPa smoke concentrations
from the Navy Aerosol Analysis and Prediction System (NAAPS).

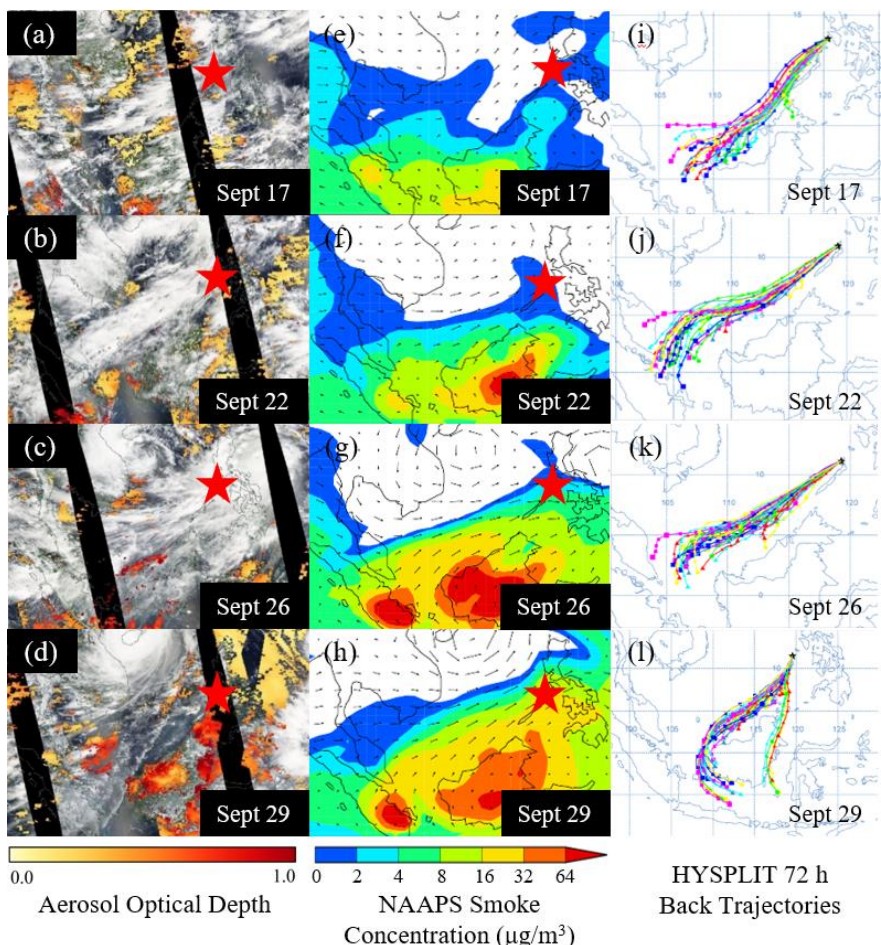


**Figure 2. Satellite images of the SCS/WPS region taken from (a-d) NASA Worldview with overlayed AOD, (e-h)**
**NAAPS smoke concentration plots (μg/m³; 850 hPa) and (i-l) HYSPLIT ensemble back trajectories during the cruise**





**(isobaric, 300m AGL, 72 hours, ending at 00:00 UTC/08:00 LST) for 18, 22, 26 and 29 Sept. Red star indicates location**
**of the *Vasco*.**
**2.2. Aerosol sampling and analysis**
Size-resolved aerosol samples were collected during the cruise using a Davis-Rotating Unit for Monitoring (DRUM)
continuously sampling cascade impactor. Samples were collected with a 10 $\mu$m inlet and eight size cuts at 5, 2.5, 1.15, 0.75,
0.56, 0.34, 0.26, 0.07 $\mu$m at a 90-minute time resolution from noontime 17 September until noontime 30 September local-time.
Particles were collected on Mylar strips coated with Apiezon grease. The eight drums were rotated at a consistent rate to create
a temporal record of mass concentration (Raabe et al., 1988). X-ray fluorescence (XRF) was performed on the DRUM samples
at the Advanced Light Source (ALS) of Lawrence Berkeley National Laboratory to measure mass concentrations of 28
elements ranging from Na to Pb. In this study, data was filtered based on location notes from the cruise such that samples
collected in the vicinity of Manila Bay were excluded from the analysis. Additionally, samples during an 8-hour pump failure
that occurred on 20 September were also excluded from the dataset. In the analysis, the stages were aggregated into three
modes: coarse (1.15-10 $\mu$m), fine (0.34-1.15 $\mu$m) and ultrafine (0.07-0.34 $\mu$m) modes.
**2.3. Model and satellite data**
NOAA Hybrid Single Particle Lagrangian Integrated Trajectory Model (HYSPLIT) back trajectories (Draxler et al.,
1998, 1999) were generated throughout the cruise period to investigate locations of aerosol emission. HYSPLIT back
trajectories have been used in several studies to establish air mass source regions (Lin et al., 2007; Cohen et al., 2010a; Atwood
et al. 2012, 2017). Back trajectories were run for 72 hours for heights of 500 m and 300 m to investigate possible vertical
inhomogeneity that has been noted in other SCS/WPS papers (Atwood et al., 2012). Trajectory endpoints corresponded to
cruise coordinates. Trajectories were constrained isobarically to limit vertical wind velocity since our area of interest is surface-
level emission.
The Navy Aerosol Analysis and Prediction System (NAAPS) reanalysis product (Lynch et al., 2016) with driving
meteorology was used to provide overall aerosol and meteorological context to the analysis.  This reanalysis utilizes a modified
version of the NAAPS as its core and assimilates quality controlled retrievals of aerosol optical depth (AOD) from Moderate
Resolution Imaging Spectroradiometer (MODIS) on Terra and Aqua and the Multi-angle Imaging SpectroRadiometer (MISR)
on Terra (Zhang et al., 2006; Hyer et al., 2011; Shi et al., 2014). NAAPS characterizes anthropogenic and biogenic fine
(including sulfate, and primary and secondary organic aerosols), dust, biomass burning smoke and sea salt aerosols. Smoke
from biomass burning is derived from near-real time satellite based thermal anomaly data to construct smoke source functions
(Reid et al., 2009), with additional orbital corrections on MODIS based emissions and regional tunings. The system has been
successfully used to monitor biomass burning plumes and to study the relationship of aerosol lifecycle to weather systems over
the MC (Reid et al., 2012, 2015, 2016; Atwood et al., 2013; Xian et al., 2013).


Active fire hotspot data was downloaded from the Fire Information for Resource Management System (FIRMS)
(https://firms.modaps.eosdis.nasa.gov/). Active fire hotspots and aerosol optical depth (AOD) at a wavelength of 550 nm were
tracked throughout the cruise via the Moderate Resolution Imaging Spectroradiometer (MODIS). MODIS detects thermal
anomalies across a region to identify possible fire activity. MODIS-derived AOD was used to derive large-scale estimates of
PM$_{2.5}$ in some studies (e.g., Zheng et al., 2017). In the study, MODIS was used to track burning emissions which were found
to be particularly prevalent in Eastern Malaysia and Indonesia. The use of MODIS to track active fire hotspots has been used
in other studies to understand seasonal trends in agricultural burning (Reid et al., 2012) and to identify and locate burning-
related sources when used in conjunction with HYSPLIT back trajectories (Atwood et al., 2017).
The NASA Worldview site (www.worldview.nasa.gov),  an application operated by the NASA/Goddard Space Flight
Center Earth Science Data and Information System (ESDIS) project, was used to supplement the satellite data by providing
true color images of the region and is particularly useful in demonstrating sudden changes of cloud environment or monsoon
flow caused by tropical cyclones.
**2.4. Positive Matrix Factorization**
Positive Matrix Factorization (PMF) was used to study the covariability of elemental species. PMF is a multivariate
factor analysis technique used in source apportionment that resolves a sample matrix $\mathbf{X}$ ($i \times j$) of $i$ samples and $j$ species into
matrices $\mathbf{F}$ ($i \times k$), $\mathbf{G}$ ($k \times j$) and $\mathbf{E}$ ($i \times j$), the source contribution matrix, source profile matrix and residual matrix,
respectively, with the assumption of $k$ factors:
$$X_{ij} = G_{ik}F_{kj} + E_{ij}$$
The goal of PMF is to determine the number of factors or sources $k$ such that the solution will be physically interpretable.
Developed by Paatero and Tapper (Paatero and Tapper, 1994), PMF is a well-established approach used in previous source
apportionment studies (Polissar et al., 1998; Lee et al., 1999; Han et al., 2006; Chan et al., 2008; Karanisiou et al., 2009; Lestari
et al., 2009; Santoso et al., 2010; Wimolwattanapun et al., 2010). PMF provides more physically realistic results compared to
other factor analysis techniques due to non-negative constraints in the model and better treatment of missing or below detection
limit (BDL) values by increasing the associated uncertainty (Paterson et al., 1999).
PMF outputs source profiles ($F$) and source contributions ($G$). PMF source profiles were normalized to the percent of
species sum, defined as the percent concentration of an element apportioned to a source. A robust mode of PMF was used for
analysis, characterized by the parameter $\alpha$. $\alpha$ is the outlier threshold distance which reduces the effect of extremely large data
points and is set at a value of 4.0 to be consistent with other PMF studies (Lee et al., 1999; Han et al., 2006).
For each of the three modes, a minimum Pearson R value of 0.0 was used to filter species based on correlations with the
mode-aggregated elemental PM concentration. This means that species negatively correlated with the summed elemental PM





concentration were not included. From the 28 species identified by XRF per DRUM stage, 20 species in the coarse mode, 22
species in the fine mode, and 19 species in the ultrafine mode were used in the size-resolved PMF. Tables S1-3 (Supplementary
Information) show the correlation coefficients of the elements, with filtered elements having negative Pearson R values against
the summed elemental PM concentration. The filtering of species through correlation was observed to improve the
interpretability of the source profiles and remove the need for the matrix rotation parameter, $F_{peak}$.

Data screening was performed based on the approach of Polissar et al. (1998) to ensure that no erroneous data points were

included in the analysis. Signal-to-noise ratios were determined and species with low ratios (less than 0.2) were excluded from
the data set (Paatero and Hopke, 2003). Below detection limit (BDL) values were replaced with half the detection limit (Han
et al., 2006). Detection limit values and error values were based on values provided by the Lawrence Berkeley National
Laboratory.

The current study employs a size-resolved PMF approach as a supplement to the other analysis methods. PMF is a

powerful tool that quantifies the contributions of PM sources and is useful for forming an initial understanding of the possible
sources from the data. However, PMF may neglect important events, particularly short-term ones, that can reveal insightful
interactions between identified sources and is unable to dissociate covarying sources as it assumes orthogonality between
factors (Van Pinxteren et al., 2016).
**3. Results I: Mass distributions and time series of selected elements**
**3.1. DRUM mass distributions**

Mass size distributions show normalized species concentrations (dM/dlogDp) across all eight DRUM stages and can be

used to validate the signal of a mode-specific tracer. In addition to isolating the signal of a tracer, changes in the mass
distributions of key elements over time indicate periods when mode-specific sources are present. Figure 3 depicts the mass
size distributions of key elements (a) potassium (K) as a tracer for biomass burning in the fine and ultrafine modes, (b) sulfur
(S), which is used as an indicator of combustion, (c) aluminum (Al) and (d) silicon (Si) which are often paired as tracers for
soil dust, (e) vanadium (V) and (f) nickel (Ni), which are often paired as tracers of oil combustion, (g) iron (Fe), another key
tracer for dust, and (h) chlorine (Cl), a reasonable tracer for sea spray given the sampling location. Figure 3 is further divided
into time periods, distinguished by color: 18-19 September (red), 19-24 September (blue), 24-27 September (green) and 27-
30 September (black).

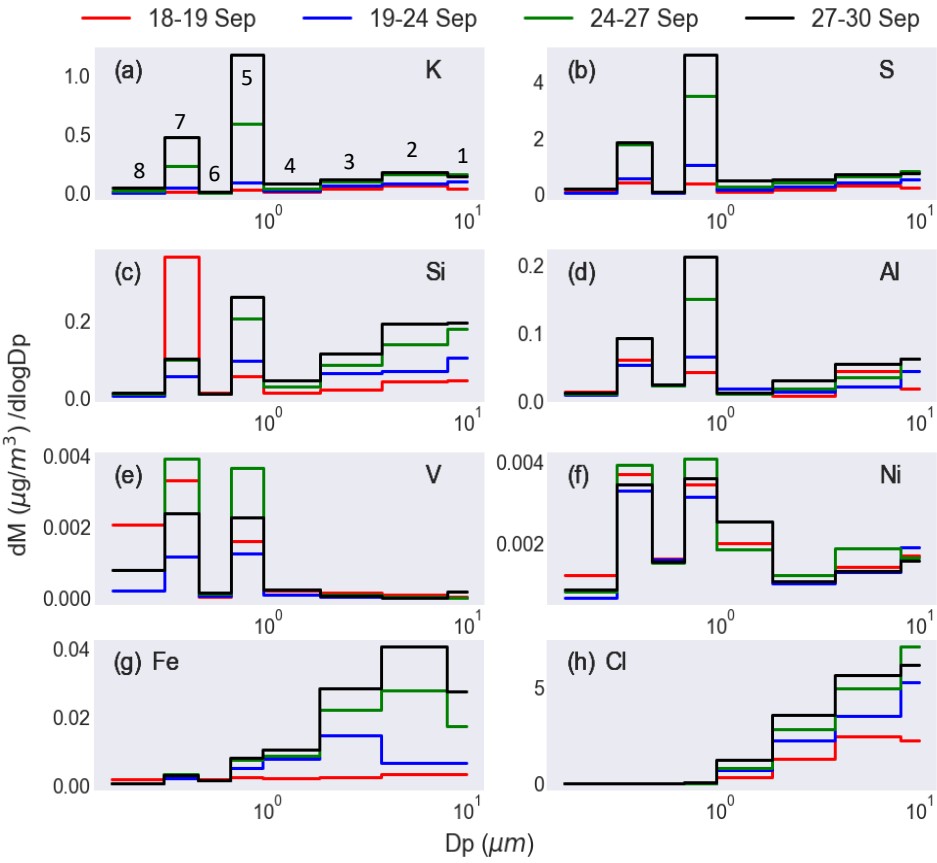

**Figure 3. Time evolution of mass size distributions of key elements over the cruise period. (a) potassium, (b) sulfur, (c) aluminum, (d) silicon, (e) vanadium, (f) nickel, (g) iron, and (h) chlorine. Time periods are colored: 18-19 Sept (red), 19-24 Sept (blue), 24-27 Sept (green), 27-30 Sept (black). Stage numbers are depicted in (a).**

Elements associated with combustion showed generally bimodal distributions with stage 5 (0.56-0.75 $\mu$m) and stage 7 (0.26-0.34 $\mu$m) peaks. K, S, Al, and Si have very similar mass size distributions over the cruise period which are suggestive of a common source (Fig. 3a-d). These elements have strong peaks in stage 5 and 7 during the whole cruise but particularly high values are observed during the last days of the sampling period (27-30 Sept). A general enhancement late in the cruise is likely related to the increase in the number of active fire hotspots reported by Reid et al. (2015), who attributed these hotspots primarily to Indonesian Kalimantan and Southern Sumatra. As the cruise took place during the end of the boreal summer, 300 m a.g.l. winds were predominantly southwesterly. A shift in back trajectories at the end of the cruise to the western and southern coasts of Borneo is observable in Fig. 2l, suggesting the source of the late-cruise enhancement to be the MC, which hosts elevated aerosol background levels from seasonal burning (Reid et al., 2013). The advection of this large aerosol event can be observed in the NAAPS smoke model over the region (Fig. 2g, h). The attribution of late-cruise aerosol enhancement to the




MC is in agreement with Reid et al. (2015) who noted that the AOD maps and southwesterly flows towards the end of the
cruise were suggestive of southwesterly transport from the MC to SCS/WPS.

Covarying behaviors of Al and Si with K and S suggest possible fine soil entrainment caught in burning updraft (Reid et

al., 2015). The stage 5 and stage 7 peaks in S are similar to those observed for northern SCS/WPS in the springtime (Atwood
et al., 2012); however, we report enhanced values, attributed to the timing of the sampling period during the MC burning
season.

Interestingly, Si shows a strong peak early in the cruise (18-19 Sept) unique to the ultrafine mode which indicates this

particular signal may not originate from soil dust but fly ash (Xie et al., 2009). As the *Vasco* was travelling past the islands of
Mindoro and Coron en route to Palawan, local sources are likely the cause of the ultrafine Si enhancement. This early-cruise
Si signal is further examined through later time series and regressions.

V shows a mass distribution characteristic of a combustion source with strong peaks in stage 5, stage 7, and stage 8 (0.07-

0.26 $\mu$m) (Fig. 3e). Almost no contribution was observed for coarser stages 1 through 4 (0.75 -10 $\mu$m), indicating that V did
not originate from soil (Lin et al., 2015) and can be treated as a tracer for oil combustion. Ni shows a similar bimodal mass
distribution (Fig. 3f) but had a larger spread over the eight stages than V, which may be due to contributions from other sources
such as fly ash (Davison et al., 1974).

Fe and Cl, well-known tracers for soil dust and sea spray, respectively, showed coarse-mode distributions that taper off

considerably in the submicron stages (Fig. 3g, h). Cl shows a purely coarse distribution which suggests sea spray considering
the sample location (Viana et al., 2008; Gugamsetty et al., 2012; Farao et al., 2014). Fe shows small peaks in stage 4 (0.75-
1.15 $\mu$m), stage 5, and stage 7; however, these do not constitute a significant signal relative to its coarse mode concentrations.
As such, we treat Fe as our coarse mode soil dust tracer. The mass distribution of Fe is observed to increase across stages 1
through 3 (2.5-10 $\mu$m) over the cruise period. The increase in coarse Fe coincides with the NAAPS-simulated transport of
smoke (Fig. 2g, h) and mirrors the enhancements of K, S, Si (Fig. 4a, b), and Al (Fig. S1a, b). These patterns suggest that
coarse soil dust accompanies smoke emissions, possibly through entrainment. The presence of soil dust is further corroborated
by Fig. 3c and Fig. 3d, which show the presence of Al and Si in the coarse mode. The distinct coarse and fine mode peaks of
Al and Si indicate separate soil dust sources. As fine mode particles have longer residence times (Cohen et al., 2010a), the fine
peaks may be an indicator of long-range transport of fine soil dust through the SCS/WPS.

Interpreting DRUM data reveals insights about the composition and interpretation of sources. Table 1 shows the ratios

of elemental $PM_{1.15}/PM_{10}$ mass concentrations. As in Atwood et al. (2012), the ratio-slope was computed by taking the slope
of the linear regression line between $PM_{1.15}$ and $PM_{10}$ mass concentrations, accompanied by $r^2$ values. Direct averages of per-
timestamp ratios of $PM_{1.15}$ and $PM_{10}$ were also taken to compute for ratio-averages, accompanied by the standard deviation of
the ratios. Fe and Cl both had ratios of 0.06, which confirm the predominantly coarse nature of these species. As commonly





used tracers of soil dust, Al and Si show moderate ratio-slope values of 0.51 and 0.29, respectively, suggesting that Al resides
in both coarse and fine ($PM_{1.15}$) modes while Si is predominantly coarse. As expected, elements commonly associated with
anthropogenic species such as V, K, and S show high $PM_{1.15}/PM_{10}$ ratios (0.8 and above) which indicate that these elemental
particles largely reside in the fine and ultrafine modes. The high ratios of V, K, and S provide evidence for the presence of
anthropogenic emissions from sources such as oil combustion and biomass burning while the low ratios of Fe and Cl support
their treatment as tracers for soil dust and sea spray, respectively.

The time-resolved DRUM data is important for showing variations in species which may be representative of

important aerosol events. Thus, observations on the time-resolved DRUM data can aid in our analysis. At the beginning of the
cruise, between 18 to 19 Sept, V, Ni, and Si show enhancements in stages 5, 7, and 8. The stage 7 Si peak during this time is
the maximum concentration over the entire cruise period, so this warrants further analysis through later time series and
regressions. The period of 19-24 Sept shows a low point in the DRUM peaks of several elements, most notably combustion
tracers K and V (Fig. 3a, c), while Cl (Fig. 3h) shows higher peaks in the coarse-mode which suggests a period of clean marine
aerosol. This period was described by Reid et al. (2015) as the cleanest of the cruise. The NAAPS model shows nearly zero
smoke concentration at the sampling site (Fig. 2f) while 72-h HYSPLIT back trajectories indicate that air masses originate
from central SCS/WPS (Fig. 2j). From 24 to 27 Sept, we observed the first major aerosol event characterized by the stage 5
and 7 enhancements of several combustion elements: K, S, Al, Si, V, and Ni (Fig. 3a-f). Fe, our coarse-mode soil dust tracer,
shows enhancements in stages 1 to 3 (Fig. 3g), which points to combustion-related entrainment of soil dust in the coarse mode.
The NAAPS model (Fig. 2g) depicts the intensification and spread of a smoke-related aerosol event that had been escalating
in southern Kalimantan since 22 Sept, reaching the *Vasco* around 26 Sept. During this mid-cruise period, concentrations of
biomass burning species K, S, Si, Al are elevated, and oil combustion tracers V and Ni show their maximum concentrations
for the cruise in stages 5 and 7 (Fig. 4e, f). The last period, 28 to 30 Sept, depicts the highest concentrations of elements
associated with biomass burning (Fig. 4a, b; Fig. S1a, b). As seen in the NAAPS smoke model (Fig. 2h) and HYSPLIT model
(Fig. 2l), the westward movement of TC Nesat across the region alters back trajectories to wind around Borneo island, reaching
southern Kalimantan which hosted a high active fire hotspot density during the time (Reid et al., 2015), thus bringing polluted
air masses toward the sampling site. Stage 5 and 7 peaks of K and S are quite notable as no other stages show significant
enhancements in response to this event. Fe, Al, and Si show similar changes but for the coarser stages 1 to 3 (Fig. 3g), indicating
a covariance of soil dust and biomass burning tracers. The temporal trends from the DRUM data serve as an entry point into
the time series analysis. By identifying key DRUM stages and time periods per element based on their mass size distributions,
we can then examine these stages to observe aerosol events over the cruise period.
**3.2. Time series of selected elements**

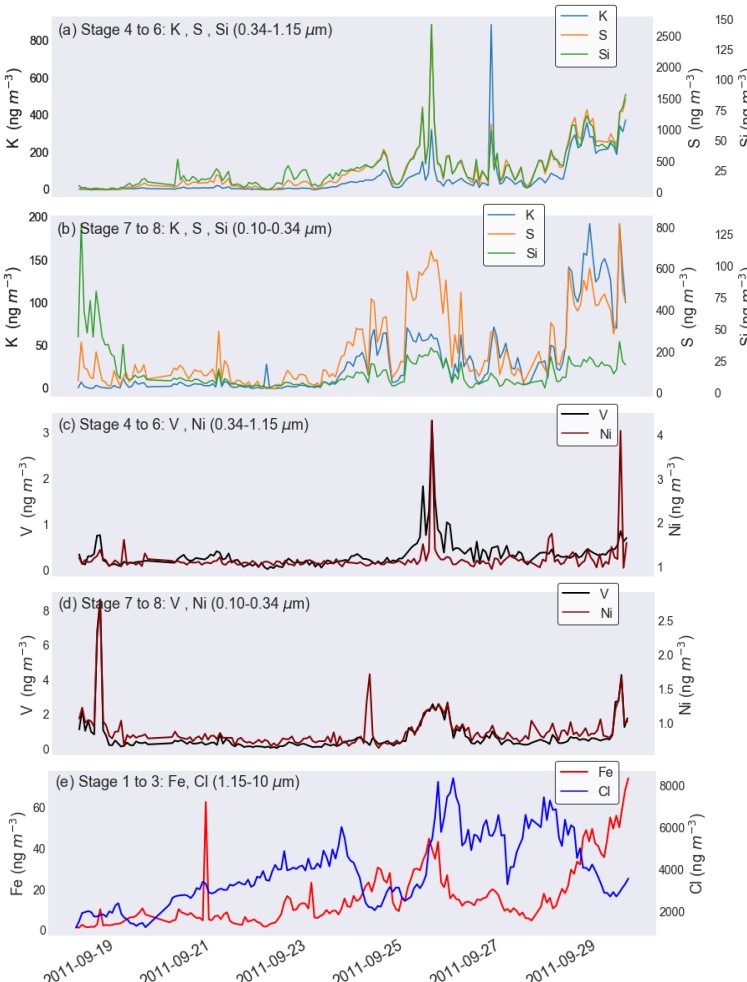

**Figure 4. Time series of (a) Stage 4-6 K, S, Si, (b) Stage 7-8 K, S, Si, (c) Stage 4-6 V, Ni, (d) Stage 7-8 V, Ni, and (e) Stage 1-3 Fe, Cl.**

The first few days of the cruise showed an 18 Sept event in oil combustion tracers V and Ni in the ultrafine mode (Fig. 4d) with a coincident but lower-magnitude response in the fine mode (Fig. 4c). Ultrafine mode V and Ni show their maxima for the cruise period during this time, expanded further in Section 5. High concentrations of ultrafine Si were sampled during this time from the beginning of the cruise until 19 Sept when it dropped to stable background levels. This early-cruise enhancement was also seen in its mass distribution plot (Fig. 3c). As the *Vasco* was traveling among islands, the Si signal may be due to local sources en route to the El Nido sampling site.

Reid et al. (2015) noted periods of clean regime after departing Manila Bay through midday 22 Sept, observable in the consistently low concentrations of various elements (Fig. 4). Chlorine shows a gradual increase in concentration from 20 Sept until 24 Sept. Chlorine, although it ages into HCl, is assumed to be fresh due to the sampling location and can therefore be used as an indicator of sea spray. Interestingly, coarse-mode Cl (Fig. 4e) showed peak concentration times during low points



in the concentrations of anthropogenic aerosol species (Fig. 4a-d), marking periods of clean marine aerosol on 22-24 Sept and 26-28 Sept, likely through wet deposition processes. During these times, back trajectories shift away from source regions and traverse open sea (Fig. 2j, k) which also hosts a lower shipping route density compared to coastal regions (Fig. S2, Supplementary Information). The first half of the cruise also saw the lowest concentrations from species associated with biomass burning, specifically submicron K, S, Si, (Fig. 4a, b), and Al (Fig. S1a, b, Supplementary Information). These species track each other quite well throughout the cruise period indicating a common source.

The event between 24 Sept and 26 Sept is observable on the time series of several key elements. The plume was the first of two distinct plume events reported by Reid et al. (2015) with the later plume occurring on 29 Sept. The enhancement of all elements in Fig. 4 suggests a mix of biomass burning, oil combustion and soil dust influences within the 24-26 Sept plume. Fine mode V and Ni show their maximum concentrations for the cruise during this event (Fig. 4c). Although these two plumes appeared as one uniform progression across the SCS/WPS region on the NAAPS smoke model (Fig. 2h), the time series showed the presence of two distinct events (Fig. 4), which is corroborated by observations from Reid et al. (2015). During this period, plume concentration dropped sharply before recovering due to the passage of squall lines sharp, observed in the time series for K, S, Si, Fe, and Cl (Fig. 4a, b, e). As concluded in Reid et al. (2015), frequent, short-term events such as this must be accounted for in studies on aerosol-convection interaction.

The period between plumes (26-28 Sept) is characterized by an overall drop in the aerosol concentration of species associated with anthropogenic sources (K, S, V, Ni; Fig. 4a-d). As Cl concentrations show peak values during this period (Fig. 4d), this indicates a period of pure marine aerosol sampling similar to the 22-24 Sept clean period. Coinciding with the passage of TC Nesat through the SCS/WPS, the observed drop in aerosol concentration is attributed to a possible restriction of shipping traffic in response to the TC and scavenging of aerosols by precipitation along the TC inflow arm (Fig. 2c) (Reid et al., 2015).

The last days of the cruise were particularly eventful as the largest aerosol event of the cruise period was visible on the NAAPS model in the form of smoke (Fig. 2h), accompanied by the spread of high AOD values throughout the SCS/WPS (Fig. 2d). Although the large areas of cloud cover created by TC Nesat hinders the detection of AOD on 26 Sept, the region is free of cloud cover by 29 Sept that significant AOD values were observed to visibly stretch from Southern Kalimantan towards the *Vasco* sampling site (Fig. 2d). In general, the NAAPS smoke transport model agrees with the spatial distribution of high AOD. Here, NAAPS modelling of smoke transport is useful in demonstrating the event's northward advection and the severity of smoke concentration in Borneo island on 26 Sept (Fig. 2h). Time series plots of elements associated with biomass burning (K, S, Si; Fig. 4a, b) and coarse mode soil dust (Fe; Fig. 4d) show significant enhancements during this time which were also observed on their mass distributions (Fig. 3). HYSPLIT back trajectories show that air masses originate from Southern Kalimantan during this period as opposed to mainland Malaysia during the first half (Fig. 2j, l). The shift in air mass trajectories is attributed to the passage of TC Nesat through the region as inflow arms from TCs have been observed to accelerate air mass





advection across the SCS/WPS, bringing more MC air into the region (Reid et al., 2012; 2015). The observed transport of
emissions from Borneo indicates that TC-enhanced long-range transport is a significant factor in SCS/WPS aerosol dispersion.

**4. Results II: positive matrix factorization and regressions**

**4.1. Source apportionment via positive matrix factorization**

To verify groupings of key elements and aid in source identification, size-resolved PMF was performed. As described in
Section 2, the eight-stage DRUM data were combined into coarse (1.15-10 $\mu$m), fine (0.34-1.15 $\mu$m) and ultrafine (0.07-0.34
$\mu$m) modes and the species included in the PMF analysis were then filtered based on their correlation to the aggregated PM
concentration. The PMF analysis resolved five sources across the three size ranges: biomass burning, oil combustion, soil dust,
sea spray and fly ash (Table 2).
Figure 5 shows the percent contribution of each source relative to the total elemental PM mass. One strength of PMF is
its quantification of a source's contribution. As expected, natural sources such as the crustal source/sea spray and soil dust
mainly contribute to the coarse mode while combustion-related sources such as biomass burning and oil combustion exist in
the fine and ultrafine modes. The existence of these sources in their expected modes is an indicator of the successful
implementation of PMF. The following sections describe the observed characteristics of sources determined by PMF.

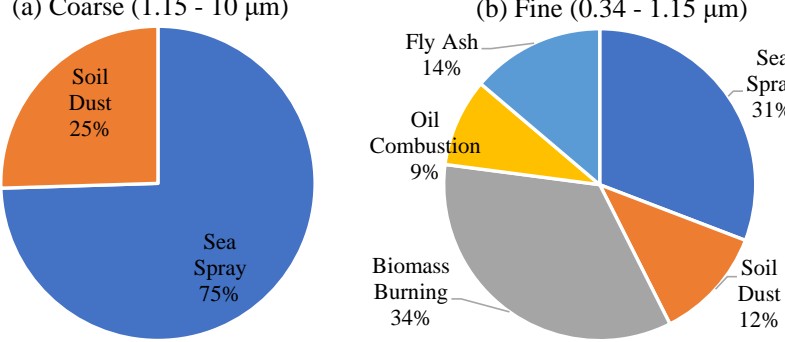

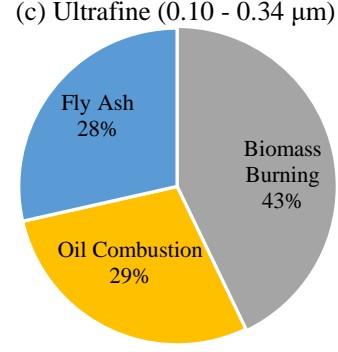

**Figure 5. Contributions of factors to the total elemental PM mass.**




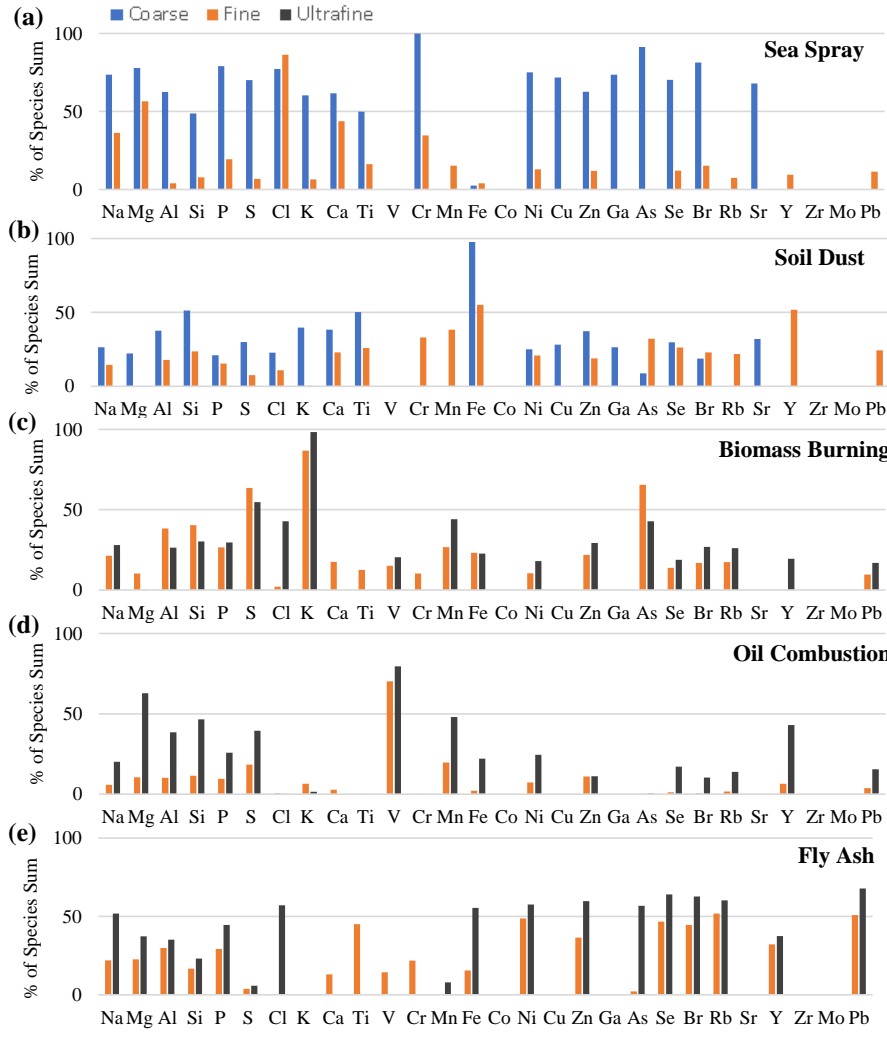


**Figure 6. PMF source profiles across different size ranges displayed by percent of species sum for (a) sea spray, (b) soil dust, (c) biomass burning, (d) oil combustion, and (e) fly ash. Coarse: Stage 1-3 (1.15-10 μm; blue), Fine: Stage 4-6 (0.34-1.15 μm; orange), Ultrafine: Stage 7-8 (0.07-0.34 μm; black).**

**Sea spray:** This factor was resolved in the coarse and fine modes characterized by strong apportionments for Na, Mg, Cl, P, and S in the coarse mode, and Cl and Mg in the fine mode as shown in the source profile of Fig. 6a. These elements are indicative of sea spray (Han et al., 2006; Wang et al., 2014). Cl has been treated as a sea spray tracer under the assumption that the sampled Cl originated from freshly produced sea spray (Atwood et al., 2012). This is likely the case for the cruise as sampling was done over sea water. The factor showed quite high mass contributions to the coarse (75%) and fine (31%) modes, attributed to the sampling location over water (Fig. 5a, b).



**Soil dust:** This factor was characterized by the presence of Fe, Al, Si, K, Ca, Ti, and Zn in the coarse mode and Fe, Cr,
Mn, and Y in the fine mode (Fig. 6b; Table 2). Several of these elements are associated with soil dust (Artaxo et al., 1990,
1998; Lestari et al., 2009; Wimolwattanapun et al., 2010; Gugamsetty et al., 2012). Soil dust may originate from the nearby
island of Palawan but also can potentially come from Borneo. The PMF model was able to distinguish between the sea spray
and soil dust factors. As sea spray aerosol is assumed to be freshly sampled during the cruise and the temporal trends of the
two sources are distinct (Fig. 7a, b), this suggests the possibility of a long-range transport mechanism for coarse mode soil
dust.  Fe serves as our tracer for soil dust due to its high apportionment in both soil dust modes. This factor showed mass
contributions of 25% and 12% in the coarse and fine modes, respectively, which indicates the predominantly coarse mode
contribution of the factor (Fig. 5a, b).
**Biomass burning:** This factor was characterized by high levels of K and S, and moderate levels of Al, As, and Si which
were found to be associated with biomass burning in previous studies (Artaxo et al., 1998; Han et al., 2006; Lestari et al., 2009;
Atwood et al., 2012; Alam et al., 2014) (Fig. 6c; Table 2). The factor showed the highest percent contributions to the PM mass:
34% and 43% in the fine and ultrafine modes, respectively. The sources of the 26 Sept and 28-30 Sept events (Fig. 7c) will be
investigated in Section 5. The presence of crustal elements Fe, Si, and Al in the source profile and the covariance of the coarse
soil dust factor (Fig. 7b) with this factor (Fig. 7c) indicate possible soil dust entrainment during burning updraft (Reid et al.,
2015; Schlosser et al., 2017).






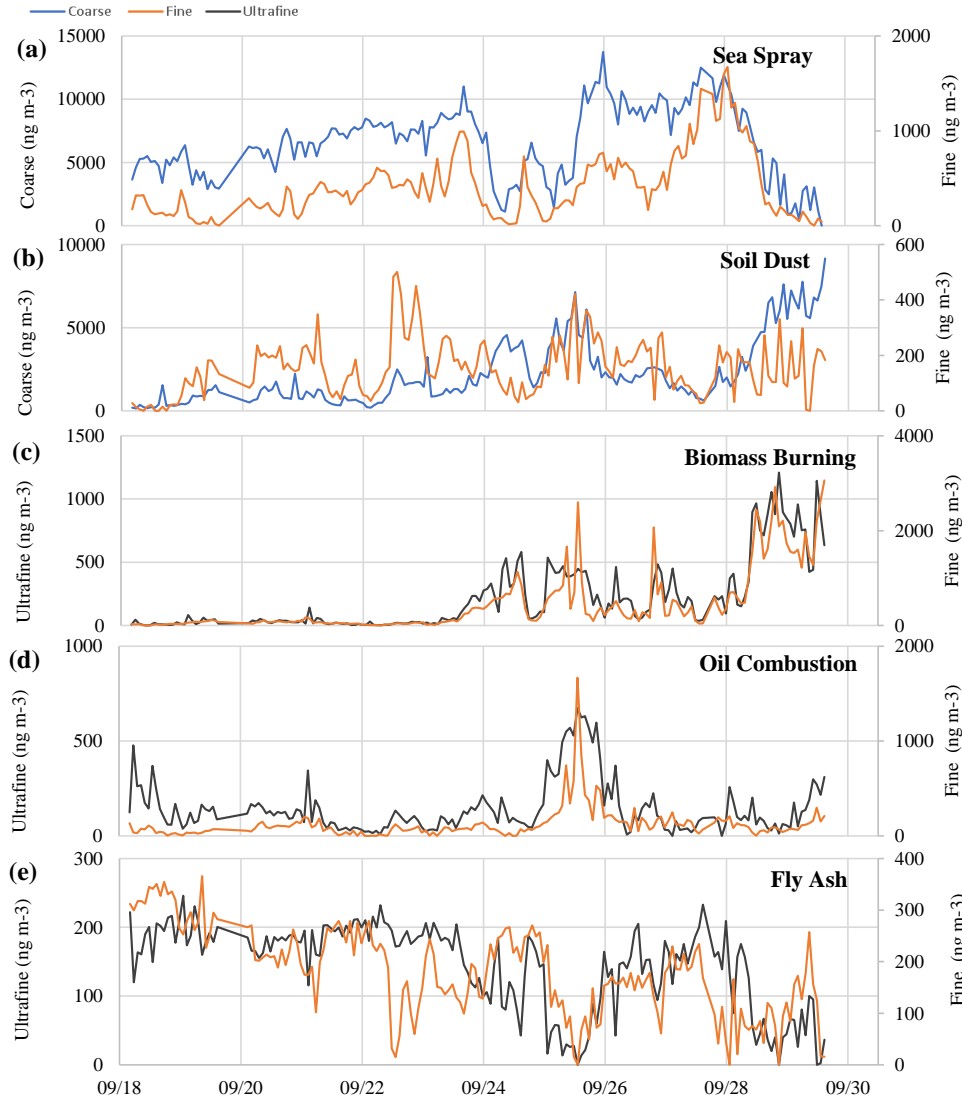

**Figure 7. PMF source contributions across size ranges displayed concentration (ng/m³) for (a) sea spray/crustal source,**

**(b) soil dust, (c) biomass burning, (d) oil combustion, and (e) fly ash.**

**Oil combustion:** This factor was characterized by high levels of V (Fig. 7d; Table 2), a well-documented tracer for oil

combustion (Hedberg et al., 2005; Mazzei et al., 2008; Becagli et al., 2012). As shown in Fig. 5, the oil combustion factor only

appeared in the fine and ultrafine sizes, contributing 9% and 29%, respectively, to the total elemental PM mass. The increasing

contribution towards finer stages corroborates the identification of the factor as an anthropogenic source. The presence of oil

combustion is expected as the SCS/WPS hosts high shipping volume, particularly in parts of the Borneo coast (Fig. S2).

**Fly ash:** This factor was observed in the fine and ultrafine modes, characterized by high levels of trace metals Ti, Ni, Zn,

Se, Br, Rb, Y, and Pb in the fine mode; Fe, Ni, Zn, As, Se, Br, Rb, and Pb in the ultrafine mode (Fig. 6e); and a source





contribution without distinct events (Fig. 7e). The presence of As, Se, Zn, and Ni are indicative of fly ash (Davison et al., 1974;
Markowski et al., 1985; Deonarine et al., 2015). The source contribution time series shows a background-type signal. The
factor contributed 14% and 28% to the total elemental PM mass for the fine and ultrafine size ranges, respectively (Fig. 5),
which is indicative of a combustion-type source. Long-range transport of fly ash from coal-fired power plants in Indonesia or
mainland Malaysia may be responsible for the appearance of the factor as no local coal-fired power plants could be found
upwind of the sampling site in 2011.

The PMF analysis resolved the presence of five sources across the ultrafine, fine and coarse modes which aids in directing

further analysis by identifying key species in the source profiles. Pearson correlation heatmaps (Fig. S3-5, Supplementary
Information) and matrices (Tables S1-S3, Supplementary Information) were constructed to examine the relationships between
species. The first column of the correlation outputs (Fig. S3-5, Tables S1-S3, Supplementary Information) shows the
correlation coefficient of the element when compared to the summed elemental PM for that mode. Similar groupings of
elements were observed when compared to the PMF source profiles, indicating the robustness of the analysis. In the coarse
mode (Fig. S3; Table S1, Supplementary Information), we observe high correlations between Na, Mg, Cl, P, S, K, Ca, Br, and
Sr, which are associated with sea spray (Han et al., 2006; Wang et al., 2014). Fe, Ti, Mn, Si, and Zn show moderate to high
correlations in the coarse mode, indicative of dust (Karanisiou et al., 2009; Wimolwattanapun et al., 2010; Lin et al., 2015;
Landis et al., 2017). In the fine mode, moderate to high correlations between Al, Si, P, S, K, Br are observed (Fig. S4, Table
S2, Supplementary Information). Several of these biomass burning elements show similarly strong correlations in the ultrafine
mode (Fig. S5, Table S3, Supplementary Information). V and Ni show a high correlation coefficient (0.91) in the ultrafine
mode, indicative of oil combustion. The excellent correspondence between the observed groupings of elements based on
correlation (Tables S2-4, Supplementary Information) and the sources resolved by PMF (Table 2) adds confidence to the
identification of key sources during the cruise. However, as PMF is an unsupervised technique, it may miss significant aerosol
events, particularly transient ones. To further expand on the relationships between elements, we turn to regression analysis.
**4.2. Regressions of selected elements**

An early-cruise ultrafine Si event was shown in the mass size distribution (Fig. 3d) and the time series (Fig. 4b) of

ultrafine Si. Fly ash was the hypothesized source of the ultrafine Si signal; however, although the PMF analysis showed the
presence of fly ash, Si was not attributed significantly to the fly ash factor (Fig. 6e). Additionally, none of the factor
contributions from PMF showed a similar trend between 18-19 Sept as ultrafine Si. Regressions show that, between 18 and 19
Sept, Si had distinct ratio slopes and the highest correlations with P ($r^2 = 0.76$), S ($r^2 = 0.73$), and Al ($r^2 = 0.61$) (Fig. 8; Table
S4) but showed poor correlations with other fly ash elements (As, Se, Pb; $r^2 < 0.12$). As it was early in the cruise, the *Vasco*
was travelling past nearby islands en route to Palawan. Therefore, local sources en route to Palawan may be the source of the
ultrafine Si enhancement.

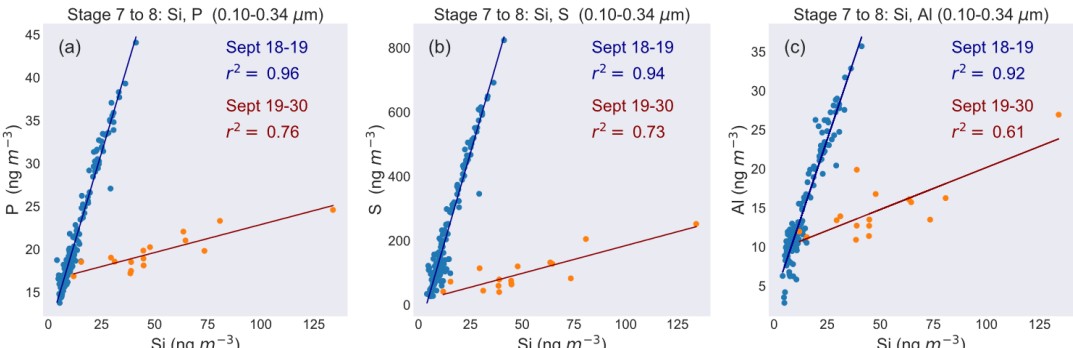

**Figure 8. Linear regressions of ultrafine Si and its most highly correlated elements (a) P, (b) S, (c) Al, divided by cruise period before Sept 19 (red) and after Sept 19 (blue).**

As S is an indicator of general combustion (Atwood et al., 2012), it is important to elucidate its relationship with tracers of other combustion sources. Fine mode and ultrafine mode linear regressions of K and V, colored by the raw concentration of S at each timestamp, were taken to show the relationships between the three species (Fig. 9a, b). S is seen to covary more with K than V as seen with the clearer color gradient following the K-axis, suggesting the origin of S during the cruise to be more dominantly from biomass burning rather than oil combustion. Multiple linear regression was also performed for these elements on the fine and ultrafine modes (Fig. S6, Supplementary Information). It was found that K and V were excellent predictors of S for most of the cruise but the model required the addition of Al to capture the variance in S between 24 and 26 Sept. A detailed description of the multiple linear regression analysis can be found in the Supplementary.

The ratio between V and Ni is often used as an indicator of the type of oil combustion source (Hedberg et al., 2005; Nigam et al., 2006; Mazzei et al., 2008; Becagli et al., 2012; Lin et al., 2015). Linear regression plots of V and Ni have a slope of 3.64 in the ultrafine mode (Fig. 9c). Nigam et al. (2006) measured a V/Ni ratio of 3.5-4 when sampling shipping emissions directly from the exhausts of various ship engines which suggests shipping to be the main source of ultrafine mode oil combustion during the cruise.

As soil composition varies geographically, soil dust ratios are excellent indicators of a plume's origin (Prospero et al., 1999; Song et al., 2006; Witt et al., 2006). Figure 9d shows linear regressions of soil dust elements in the coarse and fine modes. Al and Si, well-known indicators of dust (Viana et al., 2008; Tian et al., 2016; Landis et al., 2017), show moderate correlations with each other in the coarse and fine modes but slightly differ in ratio-slopes between the fine (Si/Al ~ 1.3; $r^2$ = 0.94) and coarse (Si/Al ~ 0.93; $r^2$ = 0.78) modes (Fig. 9d). This indicates a source of fine mode soil dust source enriched in Si; however, this could also be a matrix effect from the XRF analysis. As the *Vasco* remained near Palawan island, local dust could be the source of coarse-mode Si-enrichment; however, soil dust from Borneo is also a possibility.





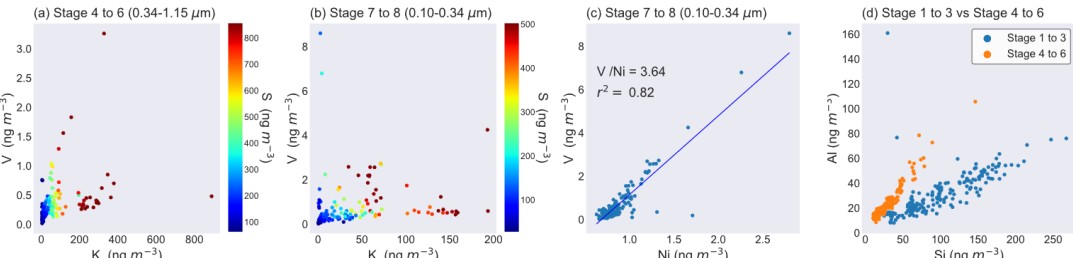


**Figure 9. Scatter plot of key species during the cruise. (a) fine mode K, V colored by the raw concentrations of S per**

**timestamp, (b) ultrafine mode K, V likewise colored by raw concentrations of S per timestamp, (c) ultrafine mode V,**
**Ni, and (d) coarse and fine mode Al, Si.**
**5. Results III: Back trajectory analysis**
**18-19 Sept: Ultrafine V, Ni enhancement from Sandakan, Sabah**
As described in Section 3, ultrafine mode V and Ni show a maximum around 18 Sept (Fig. 4d). As the *Vasco* was traveling
near local islands, the event may originate from a local source; however, back trajectories propose an oil combustion source
in Borneo. Back trajectories were generated every hour between 14:00 to 18:00 UTC (corresponding to 22:00 to 02:00 LST)
on 18 September and show a westward shift along the eastern coast of Borneo (Fig. 10a). The coast of Borneo is largely forest
(Fig. 10b) but hosts the city of Sandakan, one of Sabah's major ports (Fig. 10c, d). In addition to shipping traffic (Fig. 10d),
Sandakan contains oil depots which are a major source of industry in the area. During the westward shift of the back
trajectories, air masses pass through Sandakan at around 16:00 UTC, approximately the time of the sampled spike in V. The
shipping activity and oil depots present in this area may be responsible for the spike in oil combustion tracers, indicating the
complexity of aerosol transport in the region as small cities like Sandakan may be a source of significant spikes in aerosol.



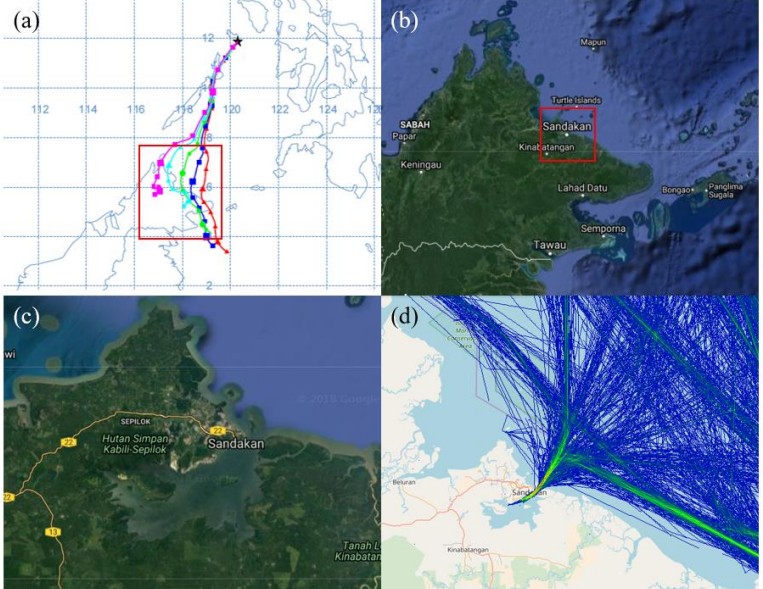


**Figure 10. Determination of 18 September event using (a) HYSPLIT back trajectories, (b, c) Google Maps view of the northeastern coast of Borneo, (d) Density of shipping traffic from Sandakan, Sabah (source: MarineTraffic). Red squares indicate the location of the succeeding plot.**

**20-24 Sept: Clean marine period**

The first half of the cruise showed the lowest concentrations of elements associated with biomass burning K, S, Si, and Al. Back trajectories during this early period originate from the northern part of Borneo and do not penetrate deeply into the MC until late into the cruise (Fig. 2l). During this period, HYSPLIT back trajectories show that air mass pathways shift away from the Borneo coasts towards open sea (Fig. 2j). In addition to the shift away from biomass burning sites, back trajectories between 22 and 24 Sept pass through areas of open sea that host lower levels of shipping traffic (Fig. S2, Supplementary Information).

**24-26 Sept: Large mixed aerosol event from northwest Borneo**

Around 26 Sept, increases in fine mode V and Ni occurred when air masses passed through the northwest coast of Borneo, suggesting the presence of ports or oil depots like with the aforementioned spike on 18 Sept from Sandakan. Back trajectories generated every 6 hours starting from 24 Sept 15:00 UTC until 26 Sept 09:00 UTC show little change over this period (not shown) and intersect with the shipping route hub located along northwest Borneo which would explain the V and Ni spikes (Fig. 2k, S1, Supplementary Information). The enrichments of biomass burning and combustion tracers K and S in the sampled air mass span a wider period beginning on 24 Sept until 26 Sept. This may be due to burning activity along the coast of Borneo which hosts several MODIS-detected active fire hotspots. Late-night land breeze from the island may have advected polluted air masses towards the coast.





**28-30 Sept: Large biomass burning event from Southern Kalimantan**
Enhancements of these elements after 28 Sept coincide with a regional increase in AOD (Fig. 2d) and are captured by
the NAAPS model in the form of a large smoke event advected northeast (Fig. 2h). Linear regressions show this large aerosol
event at the end of the cruise as a distinct group of points with enhanced concentrations of K and S (Fig. S7, Supplementary
Information), suggesting an increase in biomass burning activity during this time. Reid et al. (2015) observed a sharp increase
in the number of active fire hotspots, particularly in Sumatra and Southern Kalimantan. As discussed prior and depicted in Fig.
2, TC Nesat played a major in role in synoptic wind patterns during the cruise, causing a shift in back trajectories after 28 Sept
to the southwest coast of Borneo island. Thus, the enhancements of submicron K, S, Si and Al likely originate from biomass
burning in the MC.
**6. Summary and conclusions**
This study describes the size-resolved aerosol elemental composition of particles collected by a DRUM rotating impactor
during the 17 to 30 September 2011 M/Y *Vasco* cruise in the vicinity of the Palawan island of the Philippines. This region was
chosen due to its location as a receptor for MC aerosol sources, such as biomass burning, oil combustion and soil dust.
Meteorological conditions during the cruise were conducive to southwesterly long range transport for seasonal burning aerosol
which was observed in the concentration time series of tracers and satellite-derived AOD. Size-resolved aerosol composition
in the coarse (1.15-10 μm), fine (0.34- 1.15 μm) and ultrafine (0.07-0.34 μm) modes were used as key tracers to ascertain
source contributions. Despite the meteorological complexity of the SCS/WPS, we can gain insights into aerosol sources by
focusing on key elemental species. The time series of key elements showed distinct events on 18-19 Sept, 24-26 Sept, and 28-
30 Sept, with clean aerosol periods between events. These aerosol events served as case studies of sources in the region. While
biomass burning is indeed a key source of aerosol, other sources such as oil combustion, sea spray, fly ash, and soil dust
contribute to the chemical profile of the SCS/WPS during the southwest monsoon. Understanding these sources is key to
characterizing aerosol composition and transport in the SCS/WPS and, by extension, developing our understanding of aerosol-
cloud behavior in the region. As back trajectory analysis and aerosol chemistry showed the presence of multiple key sources,
the general conclusions of the study show that:
1.   Mass distributions of key elements showed the evolution of aerosol chemistry throughout the cruise and

interesting covariances between modes. Stage 5 (0.56-0.75 $\mu$m) and stage 7 (0.26-0.34 $\mu$m) showed enhanced

peaks in several elements associated with combustion. Throughout the cruise, mass distributions of V and Ni

track each other well both temporally and across DRUM stages, indicative of oil combustion. Mass distributions

of V and Ni show higher values in the ultrafine mode between 18-19 September, indicative of an early oil

combustion-enriched air mass which was identified to possibly originate from Sandakan, Sabah in Borneo. Mass

distributions of K, Al and S show large enhancements in the fine and ultrafine modes after 27 September





coincident with a large aerosol event reported by Reid et al. (2015). In combination with the rapid spread of high
AOD and NAAPS-modelled smoke concentration across the region, the strong peaks of these elements at the end
of the cruise provide evidence for high levels of MC burning at the end of the cruise. Coarse-mode soil dust
elements such as Fe and Si showed similarly-timed enhancements, attributed to soil particle entrainment during
burning.
2. Short-term meteorological events such as TC Nesat played a key role in long-range transport as they propagated
through the region, expediting the northeastward advection of aerosol emissions, an effect observed in previous
studies (Atwood et al., 2012; Reid et al., 2012, 2015). The sudden variations in aerosol concentration after 24
Sept can be connected to the movement of TC Nesat through the region. Prior to these events, aerosol
concentrations remained at generally low levels as NAAPS shows smoke was largely constrained to the southern
hemisphere. The passage of TC Nesat advected air masses more northward, allowing them to penetrate deep
enough into the northern hemisphere to be sampled by the *Vasco*. The TC's passage coincided with a shift in air
mass origin from mainland Malaysia prior to 24 Sept to areas known for intense burning activity, most notably
Southern Kalimantan by the end of the cruise. This corresponded to a mixed aerosol event from 24 to 26 Sept
attributed to Brunei, Borneo and a significant increase in biomass burning tracer concentrations from 28 to 30
Sept attributed to Southern Kalimantan. Between these aerosol events, a clean marine event from 26 until 28 Sept
was characterized by high concentrations of Cl and low levels of elements associated with anthropogenic sources.
Back trajectories showed that air masses travelled through the open, central SCS/WPS which suggest nearly pure
sea spray was sampled.
3. Five sources across the three modes were resolved by the PMF analysis: biomass burning, oil combustion, soil
dust, sea spray, and fly ash. A threshold Pearson R coefficient of 0.0 was used to filter species included in the
PMF analysis to improve the interpretability of the PMF solution. Results show that natural sources, sea spray
and soil dust, were observed in only the coarse and fine modes while anthropogenic sources, biomass burning,
oil combustion, and fly ash, were resolved purely in the fine and ultrafine modes. A strong correspondence
between key elements seen on the PMF source profiles and groupings of these elements on the correlation
matrices adds confidence to the PMF solution. The biomass burning PMF factor showed the highest percent
contributions to total elemental PM mass: 34% in the fine mode, and 43% in the ultrafine mode. It is interesting
to note that the contribution of the oil combustion factor increased significantly towards finer modes, 9% in the
fine mode but 29% in the ultrafine mode, corroborating its anthropogenic identification. In terms of aerosol
events, PMF source contributions were able to capture the most events seen in the raw elemental concentrations.
Differences in the temporal variations between PMF-resolved sources suggest these sources are distinct.
However, PMF did not differentiate between an early ultrafine Si spike from a distinct, subsequent spike in V

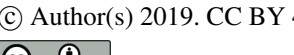



which demonstrates that PMF may merge events, leading to a loss in resolution as observed in other studies (Van
Pinxteren et al., 2016). This, however, can be ameliorated with an in-depth, supervised analysis of the data as
done in this study.
4.   As stated above, spikes in oil combustion tracers V and Ni were observed on 18 Sept in the fine and ultrafine
modes. HYSPLIT back trajectories suggest the origin of the air mass as Sandakan, an industrial area and port
city of Sabah known for its oil depots and shipping activity located along the northeastern coast of Borneo. The
spike in oil combustion suggest that a small city can cause drastic increases in tracer concentration depending on
air mass trajectories. The strong presence of ultrafine mode Si from 18-19 September was also observed but the
time series of Si is distinct from the time series of V and Ni, suggestive of a source distinct from oil combustion.
5.   The 24 to 26 September event coincided with the arrival of TC Nesat east of Luzon (northeast of the *Vasco*'s
location). Enhancements of multiple key tracers for biomass burning, oil combustion and soil dust were observed,
indicative of aerosols mixing within an air mass during transport. Biomass burning tracers K, S, Si, Al show
enhancements over a wider period (24-26 Sept) than that of oil combustion tracers V and Ni, which spiked at the
end of the period. Furthermore, aerosol-convection interactions were observed as sharp dips in the concentrations
of biomass burning and soil dust tracers around 25 Sept before recovery. Interestingly, this dip was not observed
for oil combustion tracers V, Ni. This cold pool event was reported in detail by Reid et al. (2015) and this study
further elaborated on its impact on PM of different elemental composition. This case demonstrates the effect of
short-term or high frequency phenomena on aerosol transport in the MC. HYSPLIT back trajectories show that
air masses begin to travel from the southwest MC in response to TC Nesat's inflow arm. Air masses during the
24-26 September event pass through Brunei, a shipping hub located along the northeastern coast of Borneo,
which explains the increase in oil combustion tracers V and Ni. The coast was also observed to host a number of
active fire hotspots. Land breeze may lead to the addition of burning plumes into the traveling air mass which
would explain the enrichment.
6.   The 28-30 September aerosol event showed an enrichment in K and S that coincided with a shift in back trajectory
origin to Southern Kalimantan, which hosts a high fire hotspot density. MC burning may be characterized by an
elevated K/S ratio and strong fine and ultrafine mode peaks in the mass distributions of S and K. The 28-30
September event also coincided with the enhancement of soil dust elements in the coarse mode, indicative of soil
particle entrainment during burning activity (Reid et al., 2015).
The study identified source locations of aerosol and characterized the plumes during the *Vasco* 2011 cruise; however,
unanswered questions remain such as the origin of the strong ultrafine Si signal detected early in the cruise (18-19 Sept) which
may be connected to a rapid local nucleation event. The source location of the PMF-resolved fly ash factor also remains
unidentified due to its complicated source contribution time series and unclear elemental profile. Investigation into cloud nuclei


(CN) properties during the cruise may be done to further validate the intensity and timing of plumes. In addition to the findings
of this study on the elemental PM, future research on other species collected during the 2011 and 2012 *Vasco* campaigns such
as trace gases may compliment and deepen our current understanding of the aerosol environment in the SCS/WPS by adding
more degrees of freedom, specifically the lifetimes of trace gases and potential for secondary aerosol formation during
transport.
**Author contribution**
MRAH performed the analysis and prepared the manuscript. MTC supervised the analysis, especially for the PMF
section. MOLC supervised the analysis and provided input for the manuscript. JSR collected the data onboard the *Vasco*,
supervised the analysis, provided input for the manuscript. PX provided the NAAPS Smoke model outputs for Fig. 2 and
provided input for the manuscript. JBS, NDL, SNYU collected the data onboard the *Vasco*. SC, YJZ performed the XRF
analysis on the data.
**Data availability**
The Vasco ship data is available through correspondence with Jeffrey S. Reid, jeffrey.reid@nrlmry.navy.mil. MODIS
AOD images were obtained from the NASA Worldview application: https://worldview.earthdata.nasa.gov/. HYSPLIT data is
accessible through the NOAA READY website (http://www.ready.noaa.gov). NAAPS aerosol reanalysis data can be accessed
at the US GODAE server: http://www.usgodae.org/.
**Competing Interests**
The authors declare that they have no conflict of interest.
**Acknowledgements**
We acknowledge the use of imagery from the NASA Worldview application (https://worldview.earthdata.nasa.gov/),
part of the NASA Earth Observing System Data and Information System (EOSDIS). The authors gratefully acknowledge the
NOAA Air Resources Laboratory (ARL) for the provision of the HYSPLIT transport and dispersion model and/or READY
website (http://www.ready.noaa.gov) used in this publication.

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





**Table 1. PM1.15/PM10 ratio slopes for elements ordered by ratio-slope.**

|  | Ratio slope | R-squared correlation | Ratio average | Standard deviation |
|---|---|---|---|---|
| V | 0.94 | 0.99 | 0.95 | 0.07 |
| K | 0.82 | 0.94 | 0.35 | 0.21 |
| S | 0.8 | 0.92 | 0.49 | 0.17 |
| Zn | 0.74 | 0.94 | 0.62 | 0.04 |
| Y | 0.7 | 0.7 | 0.53 | 0.11 |
| Zr | 0.7 | 0.63 | 0.65 | 0.07 |
| Mo | 0.7 | 0.67 | 0.65 | 0.04 |
| Ti | 0.68 | 0.7 | 0.53 | 0.08 |
| Rb | 0.61 | 0.64 | 0.73 | 0.09 |
| Al | 0.51 | 0.68 | 0.55 | 0.12 |
| Pb | 0.47 | 0.44 | 0.67 | 0.06 |
| Cu | 0.4 | 0.42 | 0.63 | 0.05 |
| Ni | 0.31 | 0.33 | 0.61 | 0.08 |
| As | 0.31 | 0.36 | 0.33 | 0.26 |
| Mn | 0.3 | 0.62 | 0.49 | 0.19 |
| Si | 0.29 | 0.56 | 0.32 | 0.13 |
| Se | 0.2 | 0.24 | 0.59 | 0.06 |
| P | 0.19 | 0.32 | 0.27 | 0.08 |
| Na | 0.16 | 0.57 | 0.17 | 0.03 |
| Sr | 0.16 | 0.11 | 0.49 | 0.08 |
| Br | 0.13 | 0.17 | 0.47 | 0.08 |
| Ca | 0.07 | 0.59 | 0.1 | 0.05 |
| Cl | 0.06 | 0.67 | 0.04 | 0.02 |
| Fe | 0.06 | 0.38 | 0.24 | 0.12 |
| Mg | 0.03 | 0.29 | 0.07 | 0.03 |
| Co | 0.03 | 0.03 | 0.57 | 0.1 |
| Ga | 0.03 | 0.04 | 0.56 | 0.09 |
| Cr | 0.01 | 0.02 | 0.19 | 0.19 |






**Table 2. Sources identified in each size range with PMF. Coarse (1.15-10 $\mu$m), fine (0.34-1.15 $\mu$m) and ultrafine (0.07-**
**0.34 $\mu$m).**

| Source | Major Components | Coarse | Fine | Ultrafine |
|---|---|---|---|---|
| Biomass Burning | K, S, Si, Al, As | | + | + |
| Oil Combustion | V | | + | + |
| Sea Spray | Cl, Mg, P, Br | + | + | |
| Soil Dust | Fe, Al, Si, Ca, Ti, Zn | + | + | |
| Fly ash | As, Se, Pb, Zn, Ti | | + | + |

