# Peer review of "Investigating size-segregated sources of elemental composition of particulate matter in the South China Sea during the 2011 *Vasco* Cruise"

_Atmospheric Chemistry and Physics, 2019_

## Referee Comment (RC1) · Anonymous Referee #1 · 2 Oct 2019

The manuscript provides detailed elemental analysis of the samples from a two week-long Vasco cruise to the remote SCS/WPS environment. It also gives some insights into aerosol sources contributing the region's aerosol load. Contributions from soil dust, oil combustion, sea spray and fly ash were established in addition to dominant biomass burning. Paper provides some insights into sources; however, the main drawback is the lack of quantitative information on the contribution from the analysed/selected elements to a total aerosol mass or number. How much mass was reconstructed with this elemental analysis and how important it is in the total mass balance? Authors refers to

cloud formation effects, however, there is no information on number concentrations or any relation of the sources and possible cloud formation.

Source contribution derived from PMF represents only the elemental part of the PM, but, as I wrote above, no quantitative information is provided. Therefore, the main conclusion from this manuscript is very qualitative and just points to an existence of other than biomass burning sources. The main question then is whether this information is new without any quantitative assessment? Also, the text on Line 22 refers to 28 selected elements, what were the criteria for selection? Figure 5 shows relative contributions to only element mass, but main components such as OM and EC are not included, not even sulphate or nitrate.

Lines 498-500: 'Understanding these sources is key to characterizing aerosol composition and transport in the SCS/WPS and, by extension, developing our understanding of aerosol- cloud behavior in the region.' Indeed, but for this you need to include quantitative mass composition information (including OM, SO4, NO3) or/and number distributions.

Another problem is mass size distributions provided in the manuscript. Very narrow modes at stages 7 and 5 (260-340 nm and 560-750 nm), but no mass at stage 6, raise many questions. How is this representative of ambient accumulation mode? Can these narrow modes be real? It seems to me, that there was a problem with stage 6, either in sampling or other processes, where accumulation mode peak should have occurred, but the mass is missing there for almost all elements. To conclude, the statement on bimodal distribution is very far from reality and the two narrow modes observed here are never observed with any online size measuring instrumentation.

Some elements are attributed to same source origin, but time evolution is different, e.g. Lines 221-222 claim 'K, S, Al, and Si have very similar mass size distributions over the cruise period which are suggestive of 221 a common source (Fig. 3a-d).', but size distributions change form red to green or blue periods is very different for Si and

K, also Al and K. Similarly, for other cases, e.g. the time trend of coarse Cl is quite different from fine, how does this agree with fine and coarse Cl coming from the same source, sea spray? Table 1 shows quite low correlation between PM 2.5 and PM10 for Cl, which is strange for the element form single sea spray source. How do you explain large contributions from other than sea salt origin elements in coarse sea spray factor (Figure 6 a)). Would that point to Cl also originating from other than sea spray sources? Or text on Line 336 claims that Si is originated from different source, which is in contradiction to the text above it.

Specific comments:

Lines 36-38: repeating the abstract,

Lines 38-48: rewrite to have normal text flow, now it is just a collection of sentences without any strong link.

Line 60: 'Soil dust and coarse mode biological particles may also play a role in as ice nuclei (O'Sullivan et 60 al., 2014), as biomass burning plumes are known to entrain such particles (Reid et al., 1998; 2005; Schlosser et al., 2017).' Elaborate on what you mean by biological particles from biomass burning.

Line 62: Sources do not mixt, aerosol particles do, similarly on line 66, sources do not have complicated chemistry and interactions.

Line 89: PMF was performed on selected PM elements/tracers, not total PM.

Line 183: how much of the total mass was the sum of the species?

Lines 189-190. Why only selected species and not all that were measured? What were the criteria for selection?

190-192. The method is not clear here. What was done?

Line 196: What do you mean by 'Below detection limit (BDL) values were replaced with half the detection limit (Han 196 et al., 2006).'

Figure 3: add more ticks to y axis. It is difficult to read now. Missing mass at stage 6 is unrealistic in ambient terms.

Lines 220-221: what is sigma of such narrow modes?

Line 258: add 'element' to mass concentrations in 'PM1.15 and PM10 mass concentrations';

Line 305: What do you wanted to say by this 'likely through wet deposition processes', elaborate.

Lines 409-410: elaborate on what you mean by 'However, as PMF is an unsupervised technique, it may miss significant aerosol events, particularly transient ones' what do you base your statement that PMF can miss events on? Reference?

Line 535: how much of the total mass is these 34

---

## Referee Comment (RC2) · Anonymous Referee #2 · 8 Oct 2019

Review of Hilario et al. paper entitled "Investigating size-segregated sources of elemental composition of particulate matter in the South China Sea during the 2011 Vasco Cruise" submitted to Atmos. Chem. Phys.

The paper summarizes an aerosol source apportionment in the South China Sea during a relatively short ship cruise. The results provide additional evidence on various sources and their variability in the region. The results are interesting and self-consistent although not very novel. Overall, I suggest the paper to be accepted in ACP after considering my comments and suggestions below.

Major comment:

1) Mass concentration: What are the typical mass size distributions for the total mass? This would be helpful to show at the beginning. 2) Why is there such a drastic difference between stages 5,6 and 7? Is this a sampling artefact or real? Is there mass on the stage 6, which seems to be drastically lower in many of the mass distribution of specific elementals (Figure 2).

Minor / technical comments:

L 17-18: On what grounds the area has the most complex aerosol-meteorological system?

L 32-34: The source analysis of aerosol mass and tracers is very far from understanding the regional aerosol-cloud interactions.

L 72, L107, L 225: boreal summer monsoon? Northern hemisphere summer? The term boreal is very specific in my field connecting to a specific vegetation type, which I think is very far from the environment pertinent to this paper.

L 73: Although MC is defined earlier, the message would be much clearer, if the name of the area would be spelled out.

L 76: Please include year for the date of the typhoon as well.

L 128: Figure 2: AGL and UTC/LST not defined in the text.

L 150: Driving meteorology from a model or reanalysis? Please specify.

L 160: MODIS is spelled out here for the first time.

L 187: Does the modes here refer to size distribution or to source specific modes? Please clarify. How much do you lose data due to filtering? This could also be an indication of some problems with sampling.

L 226: a.g.l. was earlier AGL. Please be consistent.

L 267-L289: A good discussion. Why is the concentration on stage 6 so much lower than on the steges 5 and 7?

L 315-316: Please correct the sentence: During this period, plume concentration dropped sharply before recovering due to the passage of squall lines sharp, . . . L 318: What do you mean by aerosol-convection interactions? How do you connect elementary composition to these interactions?

L 337: Section 4 is well written and informative.

L 380: Figure 7: I would show the mass concentrations in units ug m-3. How does these numbers correspond to the integrated mass concentrations from the filters (total mass)?

L 411: Please summarize the results of the regression results at the end of the section. In the current form it is difficult to see the importance of the findings (Sulfur and connection to biomass burning, V/Ni ratio in connection with oil combustion and discussion on Si-enhancement). Maybe a reorganization with Sect 5 would help to convey the message? In the current form, Sect 5 is very short and it could be integrated with the earlier section.

L 430: Supplementary material.

L 444: What do you mean by "timestamp"? A specific concentration at a given time?

L502: See my comment on the stage 6. Is it feasible to have such a drastic difference between three adjacent size ranges (stages 5,6,7)?

L 511: . . . evidence of high levels of MC burning? Please clarify.

L 514: Please remind the reader that TC is a tropical cyclone.

L 528: three size modes

L 571: Rapid nucleation event is brought up only in the conclusions. Also secondary

formation during transport is brought up at the end. Please clarify. Is there data to support this?

---

## Author Comment (AC1) · 19 Nov 2019

Tuukka Petäjä, PhD

Handling Co-Editor

Atmospheric Chemistry and Physics

19 November 2019

Dear Dr. Tuukka Petäjä,

[Figure]

Please find the revised version of the manuscript, entitled "Investigating size-segregated sources of elemental composition of particulate matter in the South China Sea during the 2011 Vasco Cruise" (acp-2019-352, initially submitted on 12 April 2019), for consideration for publication as a research article in Atmospheric Chemistry and Physics.

The authors express their appreciation for the two reviewers and the handling co-editor. We believe that your feedback has improved the quality and clarity of the manuscript. In the following pages, we provide point-by-point replies to the reviewer comments and questions regarding the original manuscript.

While addressing the reviewer comments on the positive matrix factorization (PMF) results, we reviewed our PMF analysis, and found that the uncertainty values should be adjusted for specie concentrations below the detection limit of x-ray fluorescence, following the approach of Han et al. (2006). This was discussed in lines 208-213 in the revised manuscript. There are no changes in the determined sources in the fine and ultrafine modes. However, the revised approach enabled us to further constrain the PMF solution and led to the identification of a third factor in the coarse mode. These coarse mode factors are soil dust, the crustal-marine mixed source, with the addition of fly ash. This change significantly improved the interpretability of the PMF source profiles by re-apportioning heavy metal elements such as Ni, Se, and Pb from the crustal-marine mixed source into its own factor (Fig. 6). The adjusted solution also improved the correspondence between the DRUM sampler and the PMF factors. Comparing the PMF solution and DRUM samples via linear regression, we report higher correlations and a linear regression slope closer to its ideal value of 1.0. We believe that this adjustment to the PMF methodology has improved our results. We have modified the appropriate figures and text in the manuscript to reflect the updated PMF results.

We thank you again for your feedback and handling of the manuscript and we trust that the latest version is now ready for publication in Atmospheric Chemistry and Physics.

Sincerely,

Miguel Ricardo A. Hilario

Corresponding Author

Manila Observatory

Ateneo de Manila University campus, Quezon City, Philippines

miguel.hilario@obf.ateneo.edu

Authors' Response to Reviewers' Comments:

Reviewer 1

General comments:

1) Paper provides some insights into sources; however, the main drawback is the lack of quantitative information on the contribution from the analysed/selected elements to a total aerosol mass or number. How much mass was reconstructed with this elemental analysis and how important it is in the total mass balance?

Thank you for your suggestion to perform mass reconstruction. To provide a more quantitative analysis, we have performed mass reconstruction using the chemically speciated PM2.5 filter substrates collected during the cruise. We note that only eight quartz and eight Teflon filters were collected for the entire two-week field campaign. We found that the PM2.5 reconstructed mass, of which elements are an important part, make up 53% of the total gravimetric mass. The result is shown in Fig. S1 of the supplementary material. A brief description is provided on lines 228-231 of the revised manuscript.

We estimated the elemental contribution to the total PM2.5 mass as the summed contributions of the sulfate, sea salt, and soil components according to formulas from Malm and Hand (2007) and Chow et al. (2015). Reconstructed elemental components derived from the DRUM sampler compose 21.2% of the total PM2.5 gravimetric mass. This is approximately twice the value calculated with the filter-based sum of the sulfate, sea salt, and soil components (11.7%). PM2.5 Teflon filters have been observed to show lower concentrations than rotating drum impactors for several elements, attributed to insufficient background subtractions (Venecek et al., 2016). A brief discussion is provided on lines 231-238 of the revised manuscript.

We are aware that elemental PM collected by the DRUM does not compose a dominant portion of the total mass; however, we do maintain that, due to their high temporal resolution (174 timestamps, 90-minute resolution), the DRUM elements serve as excellent tracers to identify sources and demonstrate source variability in the region.

Having only eight sets of filter substrates from the cruise, we chose to perform the PMF analysis on the elements from the DRUM sampler due to the higher temporal resolution and additional degrees of freedom through size-resolved measurements (eight stages from 0.10 $\mu$m to 10 $\mu$m). This enabled us to identify sources and to highlight the source variability present in the South China Sea region. We have added this explanation of an elements-only PMF analysis to lines 219-225 of the revised manuscript.

2) Source contribution derived from PMF represents only the elemental part of the PM, but, as I wrote above, no quantitative information is provided. Therefore, the main conclusion from this manuscript is very qualitative and just points to an existence of other than biomass burning sources. The main question then is whether this information is new without any quantitative assessment?

The authors agree with the relevant concern of the reviewer that the PMF-calculated source contribution represents only the elemental part of the PM. Although filter data was collected to measure organic and black carbon, only eight filters were collected in the two weeks and only cover the PM2.5 size range. Thus, as stated above, we chose to perform the PMF analysis on the elemental PM data because of its higher temporal resolution and size segregation.

While the two week-long intensive research cruise does not allow for a fully quantitative inventory of sources, the dataset presented a unique opportunity to identify aerosol sources and investigate their variability brought by the temporal nature of emissions and the complex meteorology in the region. The research cruise provided the first-ever ship-based measurements of near-surface aerosols in the South China Sea near the Philippines (Reid et al., 2015). Furthermore, these measurements enabled us to test the long-range transport hypothesis proposed in Reid et al. (2012) and provide validation for previous modeling studies on regional aerosol transport (Xian et al. 2013). We have added a short paragraph on this relevant concern in lines 219-225 of the revised manuscript.

3) Also, the text on Line 22 refers to 28 selected elements, what were the criteria for selection?

The "selected elements" refer to the elements measurable by XRF, which range from Na to Pb. We have revised the statement to remove the ambiguity. It now reads as follows "Size-segregated aerosol data was collected using a Davis Rotating-drum Unit size-cut Monitor sampler and analyzed for concentrations of 28 elements measured via X-ray fluorescence (XRF)" on lines 21-23 of the revised manuscript.

4) Figure 5 shows relative contributions to only element mass, but main components such as OM and EC are not included, not even sulphate or nitrate. Lines 498-500: 'Understanding these sources is key to characterizing aerosol composition and transport in the SCS/WPS and, by extension, developing our understanding of aerosol- cloud behavior in the region.' Indeed, but for this you need to include quantitative mass composition information (including OM, SO4, NO3) or/and number distributions.

The authors are aware that the current characterization does not include OM and EC, which are important aerosol species in an aerosol environment influenced by biomass burning. In this study, we use potassium (K) as our main biomass burning tracer with sulfur (S) as a supporting tracer for general combustion. The ionic species, elemental

carbon, and black carbon were indeed measured on eight sets of PM2.5 filters; however, as stated above, authors favored the size-resolved, high temporal-resolution elemental data collected by the DRUM sampler as its high number of data points is more appropriate for PMF and its size-resolved collection provides an additional degree of freedom for the analysis. This is explained in lines 219-225 of the revised manuscript. An analysis of the speciated data from the filters has been conducted by Reid et al. (2015).

As mentioned above, while the two-week sampling period does not allow for an exhaustive inventory of sources, the elemental dataset enabled us to identify aerosol sources, both local and regional, and investigate the temporal nature of sources present in the South China Sea region.

5) Another problem is mass size distributions provided in the manuscript. Very narrow modes at stages 7 and 5 (260-340 nm and 560-750 nm), but no mass at stage 6, raise many questions. How is this representative of ambient accumulation mode? Can these narrow modes be real? It seems to me, that there was a problem with stage 6, either in sampling or other processes, where accumulation mode peak should have occurred, but the mass is missing there for almost all elements. To conclude, the statement on bimodal distribution is very far from reality and the two narrow modes observed here are never observed with any online size measuring instrumentation.

The sharp decrease in concentrations in stage 6 despite the high concentrations in stages 5 and 7 has been observed in other studies involving the DRUM sampler; this is likely due to DRUM sampling artifacts (Atwood et al., 2013). The authors understand that the bimodal distribution likely does not reflect the true mass of the sampled aerosol, but we are simply stating in lines 220 and 242 of the original manuscript that the DRUM sampler shows a bimodal type of distribution across its stages. To elaborate on this concern and address the low concentrations in stage 6, we have added a description on lines 143-148 of the revised manuscript, which read: "A large difference in the concentrations of stage 6 (0.34-0.56 $\mu$m) compared to adjacent stages 5 (0.56-0.75

$\mu$m) and 7 (0.26-0.34 $\mu$m) was observed. The sharp decrease in concentrations in stage 6 despite the high concentrations in stages 5 and 7 has been observed in other studies involving the DRUM sampler; this is likely due to DRUM sampling artifacts and does not reflect the true aerosol mass distribution (Atwood et al., 2013). In this study, we simply report the mass distributions as sampled by the DRUM." We have removed the term "bimodal distribution" in the revised manuscript for clarity.

6) Some elements are attributed to same source origin, but time evolution is different, e.g. Lines 221-222 claim 'K, S, Al, and Si have very similar mass size distributions over the cruise period which are suggestive of 221 a common source (Fig. 3a-d).', but size distributions change form red to green or blue periods is very different for Si and K, also Al and K. Similarly, for other cases, e.g. the time trend of coarse Cl is quite different from fine, how does this agree with fine and coarse Cl coming from the same source, sea spray? Table 1 shows quite low correlation between PM 2.5 and PM10 for Cl, which is strange for the element form single sea spray source. How do you explain large contributions from other than sea salt origin elements in coarse sea spray factor (Figure 6 a)). Would that point to Cl also originating from other than sea spray sources? Or text on Line 336 claims that Si is originated from different source, which is in contradiction to the text above it.

Thank you for the comment on the size distribution changes. We meant to say that during the last half of the cruise, a regime shift occurred when back-trajectory origins shifted to southern Kalimantan (Fig. 2). This led to enhancements in K, S, and Si, which are indicative of a common source, biomass burning. We have clarified this on lines 256-260 of the revised manuscript. We also note that Figure 3 was edited in response to Reviewer #2 to include the sum of elemental PM. To maintain the eight-plot arrangement, the authors excluded Al since the other elements already provide an excellent description of the air mass evolution during the cruise and the Al size distribution does not contain significant information that cannot be derived from the other elements in the figure (K, S, Si, Fe in Fig. 3b-d, g in the revised manuscript).

Thank you for the comment on factor identification. To better reflect its elemental composition, we have changed the name of the coarse mode factor to "Crustal-Marine Mixed Source" while keeping the fine mode factor as "Sea Spray" due to high apportionments of Cl. The mixed nature in the coarse mode is likely related to covariance between local dust from nearby islands and sea spray. We have added a description of the "Crustal-Marine Mixed Source" in the PMF section, lines 399-410 of the revised manuscript which read:

"Crustal-marine mixed source: The crustal-marine mixed source was resolved in the coarse mode and is characterized by high apportionments for Na, Mg, Cl, P, and S (Fig. 6a). This source explains nearly half of the variation in crustal elements such as Al, Si, and Ca. Na and Cl show the highest contribution to the factor mass which indicate marine influence (Fig. S4, Supplementary material). These elements are indicative of a mix of sea spray and crustal sources (Han et al., 2006; Wang et al., 2014), thus its identification as a crustal-marine mixed source. The mixed nature of the source points to the covariance of local crustal emissions from nearby islands with sea spray emissions. Cl has been treated as the tracer for this factor due to its high factor sum apportionment (Fig. S4, Supplementary material) and is considered marine in origin under the assumption that the sampled Cl originated from freshly produced sea spray (Atwood et al., 2012). This is likely the case for the cruise as sampling was done over sea water. The factor showed quite high mass contributions to the coarse mode (56.8%) indicating its dominant influence on coarse elemental PM (Fig.5a). Although both this factor and the coarse mode soil dust factor are related to crustal emissions, the crustal-marine mixed source is distinct from the coarse mode soil dust factor in terms of its temporal trend, most apparent during the 28-30 Sept aerosol event (Fig. 7a, b)."

We have clarified the identification and description of fine mode "Sea Spray" in lines 411-417 of the revised manuscript, which read: "Sea Spray: This factor was resolved in the fine mode and shows high apportionments for Na, Mg, Cl, and Ca. The identification of the factor as sea spray is evidenced by the nearly 100% source apportionment of Cl. This factor showed fine (30.4%) modes, attributed to the sampling location over water. As noted prior, the appearance of this factor in the PMF analysis is due to the persistence of Cl in the 0.75-1.15 $\mu$m of the DRUM sampler (Fig. 3h). The covariance of the sea spray factor in the fine mode with the crustal-marine mixed source in the coarse mode suggest the influence of marine aerosol to some extent in both fine and coarse modes, as suggested by a moderate correlation (0.67) between PM10 and PM2.5 Cl (Table 1)."

Specific comments:

Lines 36-38: repeating the abstract,

We have reworded the sentence to avoid redundancy. Lines 38-39 of the revised manuscript now read "In the midst of several developing countries, the South China Sea/West Philippine Sea (SCS/WPS) is a receptor for a multitude of natural and anthropogenic sources of aerosol."

Lines 38-48: rewrite to have normal text flow, now it is just a collection of sentences without any strong link.

Thank you for your suggestion; we have revised the paragraph to flow more smoothly. Lines 40-48 of the revised manuscript now read: "Thus, the SCS/WPS hosts one of the world's most complex and sensitive composition and climate regimes (Balasubramanian et al., 2003; Yusef and Francisco, 2009; Atwood et al., 2013; Reid et al., 2012, 2013, 2015). The SCS/WPS is known to be impacted not only by dust storms and industrial pollution from China (Wang et al., 2011; Atwood et al., 2012) but also by biomass burning emissions from the Maritime Continent (Balasubramanian et al., 2003; Lin et al., 2007; Cohen et al., 2010a, 2010b; Wang et al., 2011; Reid et al., 2013, 2015, 2016). The transport of such emissions is enabled by the long atmospheric residence times of fine particles (Cohen et al., 2010a), potentially creating regional and global concerns through their effects on radiative forcing (Nakajima et al., 2007; Boucher et

al., 2013; Lin et al., 2013; Ge et al., 2014) and cloud properties (Sorooshian et al., 2009; Lee et al., 2012; Boucher et al., 2013; Ross et al., 2018)."

Line 60: 'Soil dust and coarse mode biological particles may also play a role in as ice nuclei (O'Sullivan et 60 al., 2014), as biomass burning plumes are known to entrain such particles (Reid et al., 1998; 2005; Schlosser et al., 2017).' Elaborate on what you mean by biological particles from biomass burning.

Thank you for your suggestion; We have reworded the sentence for clarity on lines 61-62 of the revised manuscript which now read: "Coarse mode dust and biogenic particles may also play a role as ice nuclei (O'Sullivan et al., 2014), as biomass burning plumes are known to entrain such particles (Reid et al., 1998; 2005; Schlosser et al., 2017)."

Line 62: Sources do not mixt, aerosol particles do, similarly on line 66, sources do not have complicated chemistry and interactions.

Thank you for the correction; we have corrected the wording accordingly (lines 63, 67 of the revised manuscript). Lines 62-64 of the revised manuscript now read: "As such, a network of interacting sources exists in the region surrounding the SCS/WPS, wherein aerosol particles mix during transport and complicate source apportionment." Lines 67-68 of the revised manuscript now read: "However, the source apportionment of aerosol particles is complicated by their complex chemistry and interactions with the marine environment (Atwood et al., 2012; 2017)."

Line 89: PMF was performed on selected PM elements/tracers, not total PM.

Thank you for the correction; we have reworded the sentence accordingly. Lines 90-91 of the revised manuscript now read: "Positive Matrix Factorization (PMF) was performed on size-segregated, elemental PM to detect possible size-specific sources (Han et al., 2006; van Pinxteren et al., 2016)."

Line 183: how much of the total mass was the sum of the species?

Based on the PM2.5 mass reconstruction, the reconstructed mass, of which elements are an important part, accounted for 53% of the gravimetric mass. Due to a lack of PM10 filter data, we are unable to perform a full PM10 mass reconstruction and thus provide the PM2.5 reconstruction instead. As the filters covered the entire PM2.5 range, a PM1.15 or PM1 mass reconstruction to ascertain the source contributions from PMF to the total gravimetric mass is not possible. Though the elements by themselves do not compose a dominant portion of the total mass (Fig. S1), they are useful as tracers to identify sources and demonstrate source variability in the region.

Lines 189-190. Why only selected species and not all that were measured? What were the criteria for selection?

Elements not necessary for improving the interpretability of the PMF results were removed, following the approach in other PMF studies (Liao et al., 2019; Ma et al., 2019). PMF results were found to be more interpretable after the filtering of elements based on their correlations with the total elemental PM mass.

We have reworded the paragraph for clarity. Lines 199-205 of the revised manuscript now read: "Prior to analysis via PMF, the 28 elements measured via XRF were filtered based on their Pearson's R correlation with the total elemental PM mass per mode in order to improve the interpretability of PMF factors. A minimum Pearson's R value of 0.0 was used, which removed elements that were negatively correlated with the total elemental PM. From the 28 elements identified by XRF, 20 elements in the coarse mode, 22 elements in the fine mode, and 19 elements in the ultrafine mode were included in the PMF analysis. Comparing profiles with and without the correlation-based filtering, there was no significant change in factor interpretation. This indicates that the removed elements were unnecessary for improving the PMF results (Liao et al. 2019; Ma et al., 2019)."

Lines 190-192. The method is not clear here. What was done?

Please see the response above. We have reworded the paragraph for clarity (lines
199-205 of the revised manuscript).

Line 196: What do you mean by 'Below detection limit (BDL) values were replaced with half the detection limit (Han 196 et al., 2006).'

We meant that elemental concentrations below the XRF detection limits were replaced with the half of the detection limit following the approach of Han et al. (2006). We have reworded the paragraph for clarity. Lines 211-213 of the revised manuscript now read: "Measured elemental concentrations below the detection limit of XRF were replaced with half the detection limit and their relative uncertainties were set to 100% as done in Han et al. (2006)."

Figure 3: add more ticks to y axis. It is difficult to read now. Missing mass at stage 6 is unrealistic in ambient terms.

Thank you for your suggestion; gridlines have been added across the plot for clarity (Fig. 3). We agree that the drop in stage 6 is unrealistic in terms of mass. This is likely a sampling artifact as it has been observed in previous studies using the same type of sampler. Please see our discussion on Stage 6 as a sampling artifact in the general comments section. We added this discussion in lines 143-148 in the revised manuscript.

Lines 220-221: what is sigma of such narrow modes?

Stages 5, 6, and 7 have sigma values of 0.292, 0.499, and 0.268, respectively. We note though that the sampled bimodal distribution is likely not reflective of the true mass distribution. The low concentrations in stage 6 of the DRUM sampler has been observed in other studies involving the sampler (Atwood et al. 2013). We have provided a short discussion on the sampling artifact on lines 143-148 of the revised manuscript.

Line 258: add 'element' to mass concentrations in 'PM1.15 and PM10 mass concentrations';

Thank you for the correction. Lines 293-295 of the revised manuscript now read: "Table

1 shows the ratios of elemental PM1.15/PM10 mass concentrations. As in Atwood et al. (2013a), the ratio-slope was computed by taking the slope of the linear regression line between elemental PM1.15 and PM10 mass concentrations, accompanied by r2 values."

Line 305: What do you wanted to say by this 'likely through wet deposition processes', elaborate.

We meant that the low concentrations of anthropogenic tracers (K, S, V) are associated with wet deposition related to precipitation (Reid et al., 2015, their Fig 7d). We elaborated on the wet deposition processes for better clarity. Lines 342-344 of the revised manuscript now read: "Wet deposition processes are likely responsible for the suppressed anthropogenic aerosol concentrations as precipitation was prevalent during these periods (Reid et al., 2015). Inversely, peaks in the concentrations of anthropogenic aerosol occurred during dry periods of the cruise when precipitation was low: 24-26 Sept and 28-30 Sept."

Lines 409-410: elaborate on what you mean by 'However, as PMF is an unsupervised technique, it may miss significant aerosol events, particularly transient ones' what do you base your statement that PMF can miss events on? Reference?

Thank you for your suggestion; we meant that PMF may merge consecutive but distinct aerosol events. We have elaborated on this for better clarity. Lines 468-471 of the revised manuscript now read: "However, as PMF is an unsupervised technique, it may not sufficiently disaggregate significant, consecutive aerosol events. Visually, two distinct ultrafine events occur between 18 Sept and 19 Sept in Si (Fig. 4b) and V, Ni (Fig. 4d) which are merged by PMF in its oil combustion factor (Fig. 7d). The disproportionate enhancement of ultrafine-mode Si over V and Ni suggests a source apart from oil combustion. "

Line 535: how much of the total mass is these 34

As described earlier, the PM2.5 reconstructed mass, of which elements are an important part, make up 53% of the total gravimetric mass. Thus the elemental PMF is expected to resolve sources detectable from the known portion (53% of the total PM2.5 mass). Due to a lack of PM10 filter data, we are unable to perform a full PM10 mass reconstruction and thus provide the PM2.5 reconstruction instead. We are aware that elemental PM does not compose a dominant portion of the total mass (Fig. S1); however, we do maintain that they serve as useful tracers to identify sources and demonstrate source variability in the region.

Reviewer 2

Major comments:

1) Mass concentration: What are the typical mass size distributions for the total mass? This would be helpful to show at the beginning.

Thank you for your suggestion to include the summed elemental PM mass; we have added the total elemental mass distribution as Fig. 3a and have adjusted the figure lettering accordingly in-figure and in-text. The size distribution shows a distinct coarse mode peak apart from the submicron peaks, which indicate the presence of both biogenic and anthropogenic sources. Changes in the total mass size distribution shows that, over time, a regime-change occurred around 24 Sept during which the general back-trajectory origin shifted to southern Kalimantan (Fig. 2), bringing smoke-enriched air masses to the sampling area. We have added a description of the total mass distribution on lines 252-256 of the revised manuscript, which reads: "The mass distribution of summed elemental PM (Fig. 3a) is informative as it shows distinct peaks in the coarse and submicron ranges, pointing to a combustion or anthropogenic signal during the cruise. The total mass size distribution shows that, over time, a regime-change occurred around 24 Sept during which the general back-trajectory origin shifts to the Maritime Continent. Comparing the magnitude of the summed mass distribution to those of the key species, it is clear that S contributed a significant part of the submicron mass."

To maintain the 8-subplot figure while retaining the information on size distributions, we have removed Al from Fig. 3 (formerly Fig. 3d) since the other elements already provide an excellent description of the air mass evolution during the cruise and the Al size distribution does not contain significant information that cannot be derived from the other elements in the figure (K, S, Si, Fe in Fig. 3b-d, g in the revised manuscript).

2) Why is there such a drastic difference between stages 5,6 and 7? Is this a sampling artefact or real? Is there mass on the stage 6, which seems to be drastically lower in many of the mass distribution of specific elementals (Figure 2).

Thank you for your question. The sharp decrease in concentrations in stage 6 despite the high concentrations in stages 5 and 7 has been observed in other studies involving the DRUM sampler; this is likely due to DRUM sampling artifacts (Atwood et al., 2013). The authors understand that the bimodal distribution likely does not reflect the true mass of the sampled aerosol, but we are simply stating in lines 220 and 242 of the original manuscript that the DRUM sampler shows a bimodal type of distribution across its stages. To elaborate on this concern and address the low concentrations in stage 6, we have added a description on lines 143-148 of the revised manuscript, which read: "A large difference in the concentrations of stage 6 (0.34-0.56 $\mu$m) compared to adjacent stages 5 (0.56-0.75 $\mu$m) and 7 (0.26-0.34 $\mu$m) was observed. The sharp decrease in concentrations in stage 6 despite the high concentrations in stages 5 and 7 has been observed in other studies involving the DRUM sampler; this is likely due to DRUM sampling artifacts and does not reflect the true aerosol mass distribution (Atwood et al., 2013). In this study, we simply report the mass distributions as sampled by the DRUM." We have removed the term "bimodal distribution" in the revised manuscript for clarity.

Minor/technical comments:

L 17-18: On what grounds the area has the most complex aerosol-meteorological system?

The area hosts a number of developing countries with increasing aerosols emissions over time associated with economic development and rapid urbanization. In addition to emissions from urban activity, seasonal practices such as agricultural burning also contribute to aerosol loadings in the region. These aerosols, when subject to local and regional meteorological phenomena, produce a complex aerosol-climate system. The role of topography in aerosol transport and meteorology is also well-studied, adding a third feedback in this already-complex environment. The original statement has been reworded for clarity. Lines 17-19 of the revised manuscript now read: "A combination of several developing countries, archipelagic/peninsular terrain, a strong Asian monsoon climate, and a host of multi-scale meteorological phenomena make the SCS/WPS one of the most complex aerosol-meteorological systems in the world."

L 32-34: The source analysis of aerosol mass and tracers is very far from understanding the regional aerosol-cloud interactions.

Thank you for raising this point. We are aware that source identification does not directly improve our understanding of aerosol-cloud interactions; however, improving our understanding of regional aerosol sources is a step towards this goal as aerosol composition and size are important factors in determining aerosol impacts on cloud properties (Dusek et al., 2006). We have rephrased the statement for clarity. Lines 33-36 of the revised manuscript now read: "Identifying these sources is not only key for characterizing the chemical profile of the SCS/WPS but, by improving our picture of aerosol sources in the region, is also a step forward in developing our understanding of aerosol-meteorology feedbacks in this complex environment."

L 128: Figure 2: AGL and UTC/LST not defined in the text.

Thank you for the correction; we have added definitions in lines 126-127 of the revised manuscript which now read: "Back trajectories were run for 72 hours ending at 00:00 Coordinated Universal Time (UTC)/08:00 Local Time (LT) and constrained to isobaric,

300m above ground level (AGL)."

L 72, L107, L 225: boreal summer monsoon? Northern hemisphere summer? The term boreal is very specific in my field connecting to a specific vegetation type, which I think is very far from the environment pertinent to this paper.

Thank you for pointing that out; we have reworded "boreal summer monsoon" to "Asian summer monsoon" (lines 73, 108, 263 of the revised manuscript).

L 73: Although MC is defined earlier, the message would be much clearer, if the name of the area would be spelled out.

Thank you for your suggestion. We have spelled out MC in line 74 of the revised manuscript, which now reads: "In particular, the cruise aimed to observe that emissions from the Maritime Continent were reaching the southwest monsoon trough."

L 76: Please include year for the date of the typhoon as well.

We have revised the sentence accordingly. Line 77 of the revised manuscript now has the date of the typhoon as "26 September 2011".

L 150: Driving meteorology from a model or reanalysis? Please specify.

Thank you for the suggestion. We have elaborated on the model meteorology. Lines 161-163 of the revised manuscript now read: "The Navy Aerosol Analysis and Prediction System (NAAPS) reanalysis product (Lynch et al., 2016) with driving meteorology from the Navy Global Environmental Model (NAVGEM) was used to provide overall aerosol and meteorological context to the analysis."

L 160: MODIS is spelled out here for the first time.

We have revised the sentence accordingly to use the acronym MODIS (line 164 of the revised manuscript) and have spelled out MODIS the first time it is mentioned in the text (line 123 of the revised manuscript).

L 187: Does the modes here refer to size distribution or to source specific modes? Please clarify. How much do you lose data due to filtering? This could also be an indication of some problems with sampling.

Thank you for your question. To clarify: by modes, we meant size distribution. From filtering, we removed eight elements (out of 28) in the coarse mode, six in the fine mode, and nine in the ultrafine mode. We have revised the sentence for clarity on the filtered data and process. Lines 199-205 of the revised manuscript now read: "Prior to analysis via PMF, the 28 elements measured via XRF were filtered based on their Pearson's R correlation with the total elemental PM mass per mode in order to improve the interpretability of PMF factors. A minimum Pearson's R value of 0.0 was used, which removed elements that were negatively correlated with the total elemental PM. From the 28 elements identified by XRF, 20 elements in the coarse mode, 22 elements in the fine mode, and 19 elements in the ultrafine mode were included in the PMF analysis. Comparing profiles with and without the correlation-based filtering, there was no significant change in factor interpretation. This indicates that the removed elements were unnecessary for improving the PMF results (Liao et al. 2019; Ma et al., 2019)."

L 226: a.g.l. was earlier AGL. Please be consistent

Thank you for the correction; we have revised the sentence accordingly on line 264 of the revised manuscript.

L 267-L289: A good discussion. Why is the concentration on stage 6 so much lower than on the steges 5 and 7?

Thank you; please see our discussion on Stage 6 as a sampling artifact in the general comments section of the authors' response and on lines 143-148 of the revised manuscript, which reads: "A large difference in the concentrations of stage 6 (0.34-0.56 $\mu$m) compared to adjacent stages 5 (0.56-0.75 $\mu$m) and 7 (0.26-0.34 $\mu$m) was observed. The sharp decrease in concentrations in stage 6 despite the high concentrations in stages 5 and 7 has been observed in other studies involving the DRUM

sampler; this is likely due to DRUM sampling artifacts and does not reflect the true aerosol mass distribution (Atwood et al., 2013). In this study, we simply report the mass distributions as sampled by the DRUM."

L 315-316: Please correct the sentence: During this period, plume concentration dropped sharply before recovering due to the passage of squall lines sharp, . . .

Thank you for the correction; we have removed the word "sharp" after squall lines on line 356 of the revised manuscript.

L 318: What do you mean by aerosol-convection interactions? How do you connect elementary composition to these interactions?

We meant that frequent, short-term events such as cold pools and squall lines must be accounted for in modeling studies in order to properly capture aerosol-convection interaction. Reid et al. (2015) observed sudden changes in aerosol concentrations in response to these short-term events which are not as well studied as larger circulation types. Lines 357-358 of the revised manuscript now read: "As concluded in Reid et al. (2015), frequent, short-term events such as cold pools and squall lines must be accounted for in modeling studies in order to properly capture aerosol-convection interaction."

L 337: Section 4 is well written and informative.

Thank you.

L 380: Figure 7: I would show the mass concentrations in units ug m-3. How does these numbers correspond to the integrated mass concentrations from the filters (total mass)?

We have converted the mass concentrations to $\mu$g m-3 in Figure 7. Mass reconstruction on the PM2.5 filters showed that the reconstructed mass, of which elements are an important part, accounts for 53% of the total PM2.5 (gravimetric) mass. As the filters covered the entire PM2.5 range, a PM1.15 or PM1 mass reconstruction to ascertain

the source contributions from PMF to the total gravimetric mass is not possible.

To further elucidate on the contribution of the elemental dataset to the total PM2.5 mass, we estimated the elemental contribution to the total PM2.5 mass as the summed contributions of the reconstructed sulfate, sea salt, and soil components according to formulas from Malm and Hand (2007) and Chow et al. (2015). Reconstructed elemental components derived from the DRUM sampler compose 21.2% of the total PM2.5 gravimetric mass. This is approximately twice the value calculated with filter-collected elemental concentrations (11.7%). PM2.5 Teflon filters have been observed to show lower concentrations than rotating drum impactors for several elements, attributed to insufficient background subtractions (Venecek et al., 2016). A brief discussion is provided on lines 231-238 of the revised manuscript.

We are aware that elemental PM collected by the DRUM does not compose a dominant portion of the total mass; however, we do maintain that, due to their high temporal resolution (174 timestamps, 90-minute resolution), the DRUM elements serve as excellent tracers to identify sources and demonstrate source variability in the region.

L 411: Please summarize the results of the regression results at the end of the section. In the current form it is difficult to see the importance of the findings (Sulfur and connection to biomass burning, V/Ni ratio in connection with oil combustion and discussion on Si-enhancement). Maybe a reorganization with Sect 5 would help to convey the message? In the current form, Sect 5 is very short and it could be integrated with the earlier section.

Thank you for your suggestion. We have rephrased the section to more clearly describe the importance of the linear regression analysis and added a paragraph to summarize the results of the linear regression. Lines 511-519 of the revised manuscript now read: "The regression analysis showed an early-cruise enhancement in ultrafine Si that was merged by PMF with a V, Ni enhancement that occurred soon after, highlighting the importance of the regression analysis in addition to PMF to investigate the temporal

characteristics of sources via elemental tracers. We suggest a local source en route to the main sampling area to be the cause of the enhancement but fly ash is unlikely the source due to low correlations with its tracers As, Pb, and Se. The analysis also showed the strong associations of S with biomass burning and oil combustion; however, S was shown to covary more significantly with the former. Oil combustion was determined to originate from shipping as indicated by a V/Ni ratio within the range of that measured by a previous shipping emission study. Finally, we infer multiple sources of soil dust between the coarse and fine modes due to distinct Si-Al ratios between modes; however, we are unable to determine the exact sources due to lack of information regarding local and regional soil dust ratios."

L 430: Supplementary material.

Thank you for the correction; we have revised the sentence accordingly on line 494 of the revised manuscript.

L 444: What do you mean by "timestamp"? A specific concentration at a given time?

Yes, that is correct. Each point of the scatter plot is colored by the concentration of sulfur at that given time. We clarified this in line 494-496 of the revised manuscript, which read: "fine and ultrafine mode linear regressions of K and V, colored by the concentration of S per given time, were constructed to show the relationships between the three species (Fig. 8a, b)."

L502: See my comment on the stage 6. Is it feasible to have such a drastic difference between three adjacent size ranges (stages 5,6,7)?

Thank you for raising this point; we agree that the drop in mass at Stage 6 is likely a sampling artifact as it has been observed in previous studies using the same type of sampler and probably does not reflect the true mass distribution of the aerosol environment. Please see our discussion on Stage 6 as a sampling artifact in the general comments section and on lines 143-148 of the revised manuscript.

L 511: . . . evidence of high levels of MC burning? Please clarify.

Thank you for the suggestion; we have revised the sentence for clarity. Line 586-588 of the revised manuscript now read: "The strong peaks of these biomass burning tracers, in combination with the rapid spread of high AOD and NAAPS-modelled smoke concentration across the region, provide evidence for intensive emissions from the MC."

L 514: Please remind the reader that TC is a tropical cyclone.

Thank you for the suggestion; we have revised the sentence accordingly on line 590 of the revised manuscript.

L 528: three size modes

Thank you for the correction; we have revised the sentence accordingly on line 606 of the revised manuscript.

L 571: Rapid nucleation event is brought up only in the conclusions. Also secondary formation during transport is brought up at the end. Please clarify. Is there data to support this?

Thank you for raising this point. We meant that the sudden ultrafine silicon enhancement noted at the start of the cruise may be related to a rapid nucleation process, as even submicron dust can be an important source of cloud condensation nuclei in a marine/coastal environment (Twohy et al., 2009). We have added clarification in the results section regarding the rapid nucleation event to lines 4482-485 of the revised manuscript, which now read: "As the Vasco was travelling near islands, the source of the ultrafine Si enhancement is likely a local source en route to Palawan. The sudden enhancement may be related to a rapid nucleation event as even submicron dust can be an important source of CCN in marine/coastal environments (Twohy et al. 2009)."

By secondary formation during transport, we meant that research using trace gases collected during the same research cruise may give us an idea on the chemical transformations that occur during transport to produce secondary species. We have revised

the final paragraph for clarity on lines 651-655 of the revised manuscript, which now read: "In addition to the findings of this study on the elemental PM, future research on other species collected during the 2011 and 2012 Vasco campaigns such as trace gases may complement and deepen our current understanding of the aerosol environment in the SCS/WPS through additional degrees of freedom, specifically utilizing the lifetimes of trace gases and inferring the potential for secondary aerosol formation during transport."

References

Atwood, S. A., Reid, J. S., Kreidenweis, S. M., Cliff, S. S., Zhao, Y., Lin, N.-H., Tsay, S.-C., Chu, Y.-C. and Westphal, D. L.: Size resolved measurements of springtime aerosol particles over the northern South China Sea, Atmos. Environ., 78, 134–143, doi:10.1016/j.atmosenv.2012.11.024, 2013.

Dusek, U., Frank, G. P., Hildebrandt, L., Curtius, J., Schneider, J., Walter, S., Chand, D., Drewnick, F., Hings, S., Jung, D., Borrmann, S. and Andreae, M. O.: Size Matters More Than Chemistry for Cloud-Nucleating Ability of Aerosol Particles, Science, 312(5778), 1375–1378, doi:10.1126/science.1125261, 2006.

Liao, H.-T., Chang, J.-C., Tsai, T.-T., Tsai, S.-W., Chou, C. C.-K. and Wu, C.-F.: Vertical distribution of source apportioned PM 2.5 using particulate-bound elements and polycyclic aromatic hydrocarbons in an urban area, J Expo Sci Environ Epidemiol, 1–11, doi:10.1038/s41370-019-0153-2, 2019.

Ma, L., Dadashazar, H., Braun, R. A., MacDonald, A. B., Aghdam, M. A., Maudlin, L. C. and Sorooshian, A.: Size-resolved characteristics of water-soluble particulate elements in a coastal area: Source identification, influence of wildfires, and diurnal variability, Atmos. Environ., 206, 72–84, doi:10.1016/j.atmosenv.2019.02.045, 2019.

Reid, J. S., Lagrosas, N. D., Jonsson, H. H., Reid, E. A., Sessions, W. R., Simpas, J. B., Uy, S. N., Boyd, T. J., Atwood, S. A., Blake, D. R., Campbell, J. R., Cliff, S. S., Holben,

B. N., Holz, R. E., Hyer, E. J., Lynch, P., Meinardi, S., Posselt, D. J., Richardson, K. A., Salinas, S. V., Smirnov, A., Wang, Q., Yu, L. and Zhang, J.: Observations of the temporal variability in aerosol properties and their relationships to meteorology in the summer monsoonal South China Sea/East Sea: the scale-dependent role of monsoonal flows, the Madden–Julian Oscillation, tropical cyclones, squall lines and cold pools, Atmospheric Chem. Phys., 15(4), 1745–1768, doi:10.5194/acp-15-1745-2015, 2015.

Twohy, C. H., Kreidenweis, S. M., Eidhammer, T., Browell, E. V., Heymsfield, A. J., Bansemer, A. R., Anderson, B. E., Chen, G., Ismail, S., DeMott, P. J. and Van Den Heever, S. C.: Saharan dust particles nucleate droplets in eastern Atlantic clouds, Geophys. Res. Lett., 36(1), L01807, doi:10.1029/2008GL035846, 2009.

Venecek, M. A., Zhao, Y., Mojica, J., McDade, C. E., Green, P. G., Kleeman, M. J. and Wexler, A. S.: Characterization of the 8-stage Rotating Drum Impactor under low concentration conditions, Journal of Aerosol Science, 100, 140–154, doi:10.1016/j.jaerosci.2016.07.007, 2016.

Xian, P., Reid, J. S., Atwood, S. A., Johnson, R., Hyer, E. J., Westphal, D. L., and Sessions, W.: Smoke transport patters over the Maritime Continent, Atmos. Res., 122, 469–485, doi:10.1016/j.atmosres.2012.05.006, 2013